# Subcellular drug targeting illuminates local kinase action

**Paula J Bucko[1], Chloe K Lombard[2†], Lindsay Rathbun[3†], Irvin Garcia[1], Akansha Bhat[1], Linda Wordeman[4], F Donelson Smith[1], Dustin J Maly[2], Heidi Hehnly[3], John D Scott[1]***

[1]Department of Pharmacology, University of Washington, Seattle, United States; [2]Department of Chemistry, University of Washington, Seattle, United States; [3]Department of Biology, Syracuse University, Syracuse, United States; [4]Department of Physiology and Biophysics, University of Washington, Seattle, United States

**Abstract** Deciphering how signaling enzymes operate within discrete microenvironments is fundamental to understanding biological processes. A-kinase anchoring proteins (AKAPs) restrict the range of action of protein kinases within intracellular compartments. We exploited the AKAP targeting concept to create genetically encoded platforms that restrain kinase inhibitor drugs at distinct subcellular locations. **Lo**cal **K**inase **I**nhibition (**LoKI**) allows us to ascribe organelle-specific functions to broad specificity kinases. Using chemical genetics, super resolution microscopy, and live-cell imaging we discover that centrosomal delivery of Polo-like kinase 1 (Plk1) and Aurora A (AurA) inhibitors attenuates kinase activity, produces spindle defects, and prolongs mitosis. Targeted inhibition of Plk1 in zebrafish embryos illustrates how centrosomal Plk1 underlies mitotic spindle assembly. Inhibition of kinetochore-associated pools of AurA blocks phosphorylation of microtubule-kinetochore components. This versatile precision pharmacology tool enhances investigation of local kinase biology.

*For correspondence:
scottjdw@uw.edu

†These authors contributed equally to this work

**Competing interests:** The authors declare that no competing interests exist.

## Introduction

Protein kinase inhibitor drugs are an emerging class of therapeutics for a variety of clinical indications (*Ferguson and Gray, 2018*). These small molecules are also powerful research tools that can be used to discover new aspects of kinase signaling (*Caunt et al., 2015*). While '*drugging*' individual kinases can establish their role in cellular events, this global approach cannot discriminate where or when these signaling enzymes operate inside the cell. Thus, designing pharmacological strategies that influence the spatial and temporal action of kinases is at the frontier of precision medicine.

Polo-like kinase 1 (Plk1) and Aurora A (AurA) are important regulators of cell division (*Barr et al., 2004*; *Combes et al., 2017*; *Lens et al., 2010*). Accordingly, ATP-competitive drugs that block their activity, such as BI2536 and MLN8237, ascribe functions to these kinases and are promising anticancer therapies (*Steegmaier et al., 2007*; *Tang et al., 2017*; *Lénárt et al., 2007*; *Asteriti et al., 2014*; *Manfredi et al., 2011*). However, elucidating the individual spatial and temporal actions of Plk1 and AurA remains challenging as these enzymes continually change their location and activity throughout mitosis (*Bruinsma et al., 2015*; *Joukov and De Nicolo, 2018*; *Lera et al., 2016*). As a result, global drug delivery strategies mask the unique contributions of each kinase at distinct mitotic structures. Moreover, standard drug regimens that saturate dividing cells with these compounds may increase off-target effects and toxicity (*Klaeger et al., 2017*).

Plk1 and AurA have been implicated in the control of mitotic progression (*Barr et al., 2004*; *Combes et al., 2017*; *Lens et al., 2010*). The anchoring protein Gravin/AKAP12 participates in this process by forming a macromolecular complex with these enzymes (*Hehnly et al., 2015*). A-kinase anchoring proteins (AKAPs) are scaffolding proteins that limit the scope of cell signaling events at

**eLife digest** In order for an animal cell to divide it needs to duplicate its DNA and split this genetic material equally between its daughter cells. This process, also known as mitosis, requires a number of different proteins that work together to coordinate this vital aspect of the cell's lifecycle. For example, two enzymes known as Polo-like kinase 1 (Plk1) and Aurora A (AurA) accumulate at specialized structures within the cell called centrosomes, where they receive signals from other parts of the cell to promote this process. Plk1 and AurA also operate at other locations in the cell, but it remains unclear whether these proteins have different activities at each individual location.

Current methods for measuring protein activities use drugs or genetic approaches that switch off the target protein's activities everywhere in the cell. Now, Bucko et al. have developed a new tool named LoKI that is able to direct drugs to specific locations in animal cells to switch off a target protein's activity only at these sites. The experiments showed that inhibiting the activities of Plk1 and AurA at the centrosomes led to defects that prolonged mitosis.

The new tool developed by Bucko et al. provides a means to more precisely study local events that occur in healthy and diseased cells. This will help us to understand how cancer and other diseases develop and may ultimately lead to new treatments for human patients with these conditions.

distinct cellular locations (*Scott and Pawson, 2009*; *Langeberg and Scott, 2015*; *Esseltine and Scott, 2013*). For example, anchored protein kinase A action is constrained to within 200–400 angstroms of the AKAP (*Smith et al., 2013*; *Smith et al., 2017*). This has led to the formulation of a signaling island model where catalytic activity of anchored kinases is restricted to the immediate vicinity of select substrates (*Scott and Pawson, 2009*; *Langeberg and Scott, 2015*; *Esseltine and Scott, 2013*). Likewise, anchoring of Plk1 and AurA occurs on a phosphorylated species of Gravin at Threonine 766 (*Canton et al., 2012*; *Hehnly et al., 2015*; *Colicino et al., 2018*). Consequently, loss or disruption of this scaffold abrogates Plk1 and AurA organization at centrosomes and promotes mitotic delay (*Hehnly et al., 2015*). Yet, an outstanding question that remains is exactly how do centrosome-localized pools of Gravin-anchored Plk1 and AurA coordinate mitotic signaling events.

In the present study, we first establish that Gravin is required for localizing active pools of Plk1 and AurA at mitotic centrosomes. We then develop a novel chemical-biology tool, LoKI (**Lo**cal **K**inase **I**nhibition), to probe the actions of Plk1 and AurA at defined subcellular locations. Finally, we demonstrate that local inhibition of Plk1 and AurA kinases at centrosomes and kinetochores disrupts substrate phosphorylation, spindle organization, and mitotic duration. Together, these studies decipher how activities of individual kinases at precisely defined microenvironments contribute to the global signaling events that underlie mitosis.

## Results

### Gravin loss perturbs duration of mitosis and accumulation of Plk1 and AurA at centrosomes

Gravin depletion via shRNA-mediated knockdown perturbs mitotic progression (*Hehnly et al., 2015*). Here, we test whether complete loss of Gravin also prolong mitosis. We used time-lapse video microscopy to monitor mouse embryonic fibroblasts (MEFs) from wild-type and Gravin knockout (KO) mice (*Figure 1A*, *Figure 1—figure supplement 1A*). Live-cell imaging of wild-type MEFs expressing GFP-tagged histone 2B (*Figure 1—video 1*) established a baseline mitotic duration (nuclear envelope breakdown to anaphase) as 32.6 min (*Figure 1B*). In contrast, mitosis was delayed by 11.9 min in Gravin KO cells (*Figure 1B*). These data further establish that Gravin promotes timely progression of cells through mitosis (*Gelman, 2010*).

Gravin is required for organization of Plk1 and AurA at mitotic centrosomes (*Hehnly et al., 2015*). Whether Gravin loss reduces active pools of Plk1 and AurA at this location remains unclear. To test this, we explored whether activity of centrosome-localized Plk1 and AurA is perturbed in cells lacking Gravin. We examined pT210-Plk1 and pT288-AurA immunofluorescence (measures of Plk1 and AurA activity, respectively) in HEK293 cells stably expressing a control or Gravin shRNA

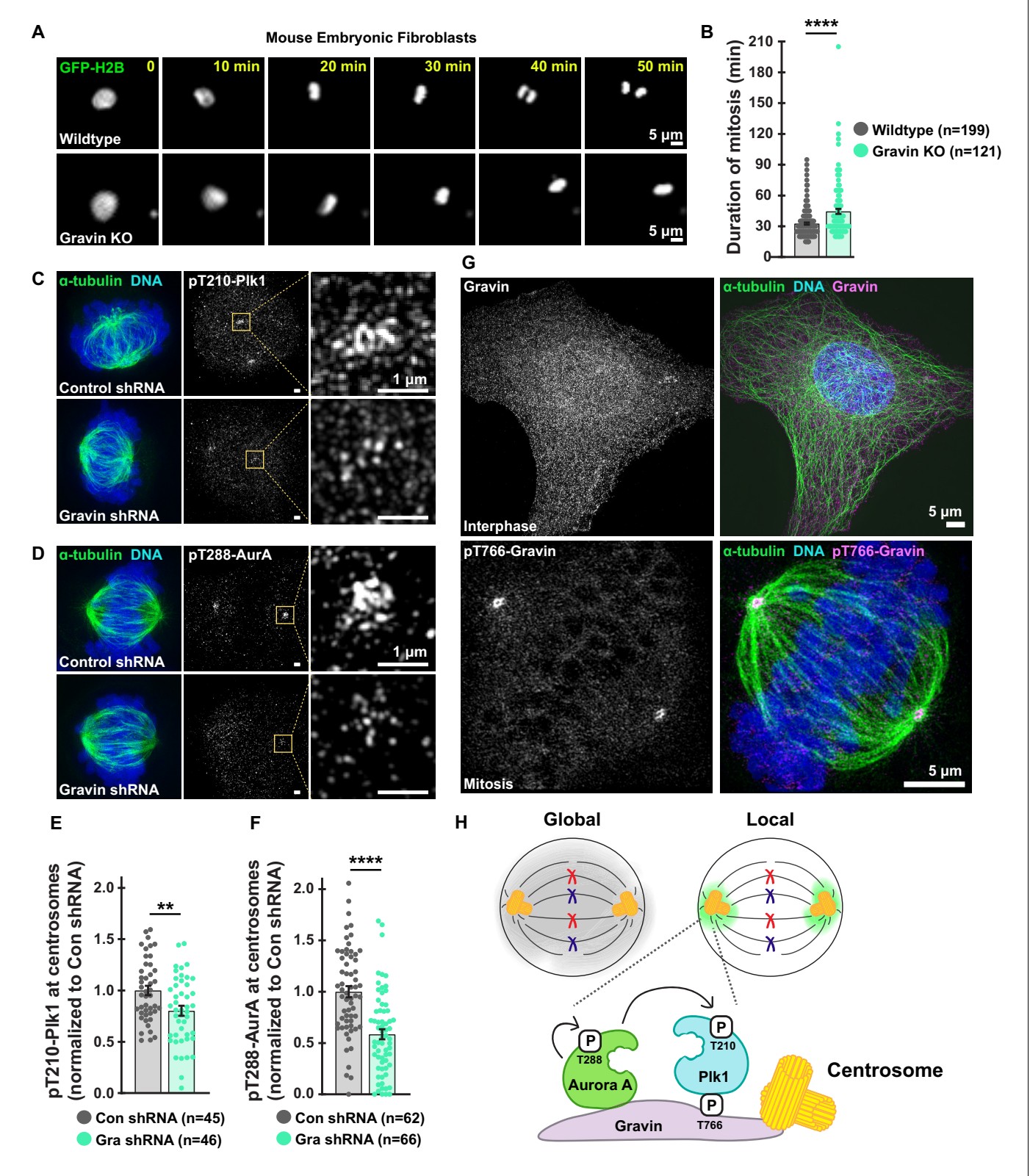

**Figure 1.** Loss of Gravin perturbs duration of mitosis and accumulation of active mitotic kinases at centrosomes. (**A**) Representative time-lapse images of primary MEFs derived from wildtype (top) and Gravin knockout (bottom) cells. Cells transiently expressing GFP-H2B were monitored through mitosis. (**B**) Quantification of time-lapse experiments depicts duration of mitosis from nuclear envelope breakdown to the onset of anaphase; Wildtype, *n* = 199, Gravin KO, *n* = 121, ****p<0.0001. (**C, D**) Structured illumination microscopy (SIM) of representative mitotic HEK293 cells stably expressing Control or

*Figure 1 continued on next page*

Figure 1 continued

Gravin shRNA. Composite images (left) depict cells stained for α-tubulin (green) and DNA (blue). Immunofluorescence of pT210-Plk1 (C) and pT288-AurA (D) as an index of kinase activity (mid) and 5X magnification of centrosomal pT210-Plk1 (C) and pT288-AurA (D) signals (insets). (E, F) Quantification of centrosomal pT210-Plk1 (E) and pT288-AurA (F) immunofluorescence. Points represent individual cells (n). Data are normalized to Con shRNA; (E) Con shRNA, $n = 45$, Gra shRNA, $n = 46$, **p=0.0036; (F) Con shRNA, $n = 62$, Gra shRNA, $n = 66$, ****p<0.0001. Experiments were conducted at least three times (N = 3) and P values were calculated by unpaired two-tailed Student's t-test. Data are mean ± s.e.m. (G) SIM micrographs of Gravin (top, gray and magenta) in interphase and pT766-Gravin (bottom, gray and magenta) in mitotic U2OS cells. Composite images (right) also depict α-tubulin (green) and DNA (blue). (H) Schematic of global drug distribution (gray) vs drug targeting to centrosomes (green). Gravin scaffolds centrosome-localized pools of Plk1 and AurA.

The online version of this article includes the following video and figure supplement(s) for figure 1:

**Figure supplement 1.** Confirmation of Gravin loss in MEFs and detection of Gravin and pT766-Gravin in mitotic and interphase U2OS cells.
**Figure 1—video 1.** Loss of Gravin perturbs duration of mitosis.
https://elifesciences.org/articles/52220#fig1video1

(*Figure 1C,D*). Gravin depletion reduced immunofluorescence of pT210-Plk1% to 80.2% of control shRNA-expressing cells (*Figure 1C,E*). Strikingly, pT288-AurA signal in Gravin-depleted cells dropped to 58.5% of control cell immunofluorescence (*Figure 1D,F*). These findings reveal that Gravin is required for localizing active pools of Plk1 and AurA at mitotic centrosomes.

## LoKI platforms direct kinase inhibitor drugs to specific subcellular locations

To decipher how Gravin-associated pools of Plk1 and AurA coordinate mitotic signaling events, we needed to selectively target drugs to this location without disrupting the Gravin-Plk1-AurA macro-molecular complex. During interphase Gravin is dispersed throughout the cell (*Figure 1G*, *Figure 1—figure supplement 1B*). In mitosis, however, pT766-Gravin accumulates at centrosomes, a major hub of Plk1 and AurA signaling (*Figure 1G*, *Figure 1—figure supplement 1B*). This provided the impetus to pharmacologically inhibit mitotic kinases at centrosomes (*Figure 1H*). A key advance in our studies came with the development of the LoKI tool which allows us to target kinase inhibitor drugs to specific subcellular locations. We fused a **p**ericentrin **A**KAP450 **c**entrosomal-**t**argeting (PACT) domain (*Gillingham and Munro, 2000*) to a SNAP-tag moiety which can be covalently labeled with chloropyrimidine (CLP)-linked substrates inside cells (*Keppler et al., 2003*) (*Figure 2A*). A CLP-conjugated analog of BI2536 (CLP-BI2536) was generated to selectively target Plk1 (*Figure 2A,B*, *Figure 2—figure supplement 1A*). In vitro kinase activity measurements demonstrated that CLP-BI2536 potently inhibits Plk1 ($IC_{50} = 49 \pm 26$ nM; *Figure 2C*, *Figure 2—figure supplement 1B*).

To generate stable cell lines, U2OS osteosarcoma cells were infected with lentiviral constructs encoding the SNAP-PACT moieties fused to an mCherry reporter (*Figure 2—figure supplement 1C*). Inducible protein expression was accomplished by a doxycycline-inducible promoter (*Figure 2—figure supplement 1D*). Immunoblot detection of mCherry-SNAP-PACT persisted up to 4 hr upon removal of doxycycline (*Figure 2—figure supplement 1E*). As anticipated, mCherry-SNAP-PACT associates with centrosomes during interphase and mitosis (*Figure 2—figure supplement 1F*). Super-resolution structured illumination (SIM) imaging revealed that the SNAP-PACT construct (magenta) was labeled by CLP-fluorescein (yellow) at centrosomes (*Figure 2D,E*, *Figure 2—video 1*). Counterstaining with α-tubulin (green) revealed the mitotic spindle and DAPI (blue) detected DNA (*Figure 2D*, *Figure 2—video 2*). Collectively these results demonstrate that centrosomal targeting of SNAP-PACT creates a platform for the delivery of CLP-conjugates (*Figure 2—figure supplement 1G*). This new drug targeting method is herein referred to as **LoKI-on** (**Lo**cal **K**inase **I**nhibition-**on**). In parallel, a LoKI-off vector containing an inactivating mutation (C144A) in SNAP-tag was constructed (*Figure 2—figure supplement 1H*). LoKI-off is unable to incorporate CLP-conjugates and serves as the control platform (*Figure 2—figure supplement 1H*).

Pulse-chase experiments were used to determine how efficiently CLP-BI2536 labeled SNAP-PACT. U2OS cells were treated with CLP-BI2536 (over a range of concentrations) to block CLP-rhodamine conjugation (*Figure 2F*). Incubation with 250 nM CLP-BI2536 for 4 hr at 37°C was defined as the optimal drug regimen (~50% labeling of SNAP-PACT; *Figure 2F*, *Figure 2—figure supplement 2A,B*). Next, we measured the pT210-Plk1 immunofluorescence signal as an index of active kinase (*Lee and Erikson, 1997*) (*Figure 2—figure supplement 3A,B*). In mitotic cells expressing LoKI-off,

incubation with 250 nM CLP-BI2536 reduced pT210-Plk1 immunofluorescence to 58.1% of DMSO-treated controls (**Figure 2G,I**). Strikingly, the pT210-Plk1 signal was reduced to 21.4% in cells expressing LoKI-on (**Figure 2H,I**). This trend persisted with lower CLP-BI2536 concentrations and even after a 1 hr washout of drug (**Figure 2I,J**). Further validation confirmed that the reduction of pT210-Plk1 does not result from a loss in total Plk1 protein at centrosomes (**Figure 2—figure supplement 3C**). Additional controls established that inducible expression of LoKI-on was necessary to attenuate the pT210-Plk1 signal (**Figure 2—figure supplement 3D**). Immunoblot analyses of noco-dazole-synchronized cells collected via mitotic shake-off further support these findings (**Figure 2—figure supplement 3E**). Parallel analyses were conducted in HeLa and hTERT-immortalized RPE retinal pigment epithelial cells (**Figure 2—figure supplement 3F–J**). We note that due to clonal cell line differences between LoKI-off and LoKI-on cells, baseline immunofluorescence signal was normalized to DMSO-treated controls (**Figure 2—figure supplement 4**). Collectively, these findings establish LoKI as a new pharmacological tool to selectively block Plk1 activity at centrosomes.

## Targeting Plk1 inhibitor drugs to centrosomes perturbs early mitotic events

Correct assembly of bipolar spindles ensures the fidelity of chromosome segregation into daughter cells (**Prosser and Pelletier, 2017**). Abrogation of Plk1 activity has been linked to mitotic spindle defects that include abnormal bipolar and monopolar structures (**Sunkel and Glover, 1988**; **Lane and Nigg, 1996**; **Sumara et al., 2004**) (**Figure 3A**). Spindle classification measurements were carried out to assess if centrosomal inhibition of Plk1 induces these morphological anomalies (**Figure 3B**). Analysis in U2OS cells revealed that application of CLP-BI2536 in LoKI-on cells increased the incidence of abnormal bipolar (green) and monopolar (purple) spindles by 24.6% as compared to LoKI-off controls (10.3%; **Figure 3B,C**, **Figure 3—figure supplement 1A**). More pronounced spindle defects were observed when local drug delivery experiments were repeated in RPE cells (**Figure 3D**, **Figure 3—figure supplement 1B**). Interestingly, local delivery of CLP-BI2536 did not further exacerbate defective spindle organization in Hela cells, which naturally exhibit a high incidence of aberrant spindles (**Figure 3E**, **Figure 3—figure supplement 1C**). Thus, targeting Plk1 inhibitor drugs to centrosomes promotes mitotic spindle defects in various cell types.

Spindle assembly relies on γ-tubulin, a protein that interacts with α/β-tubulin polymers (**Moritz et al., 1995**; **Zheng et al., 1995**). Plk1 phosphorylates pericentriolar substrates that coordinate γ-tubulin accumulation at mitotic centrosomes to facilitate microtubule nucleation (**Lane and Nigg, 1996**; **Haren et al., 2009**; **Xu and Dai, 2011**) (**Figure 3F**). Accordingly, we monitored centro-somal accumulation of γ-tubulin in U2OS cell after application of CLP-BI2536 for 4 hr followed by a 1 hr washout (**Figure 3G–I**, **Figure 3—figure supplement 1D**). The γ-tubulin signal (yellow) accumulated at centrosomes in LoKI-off controls (**Figure 3G,I**). However, γ-tubulin levels were drastically reduced when LoKI-on cells were exposed to the same drug regimen (**Figure 3H,I**). Amalgamated data from five independent experiments are presented (**Figure 3I**). These findings indicate that targeting CLP-BI2536 to centrosomes impairs accumulation of γ-tubulin at this location.

## Targeting AurA inhibitors to centrosomes suppresses AurA activity

A versatile feature of the LoKI system is the ability to compartmentalize a variety of drug analogs. The AurA inhibitor MLN8237 was a logical candidate to highlight the broad applicability of this approach. CLP-MLN8237 was synthesized (**Figure 4A**, **Figure 4—figure supplement 1A**). In vitro kinase activity measurements demonstrated that CLP-MLN8237 potently inhibits AurA (IC$_{50}$ <9.5 nM; **Figure 4B**, **Figure 4—figure supplement 1B**). Cell-based characterization established that treatment with 100 nM CLP-MLN8237 for 4 hr at 37°C was sufficient to label ~50% of drug binding sites (**Figure 4C**). Next, we used immunofluorescent detection of pT288-AurA as an index of kinase activity (**Figure 4—figure supplement 2A,B**). In mitotic cells expressing LoKI-off, incubation with 100 nM CLP-MLN8237 reduced pT288-AurA immunofluorescence to 27.8% of DMSO-treated controls (**Figure 4—figure supplement 2C**). Importantly, the pT288-AurA signal was further reduced to 14.2% in cells expressing LoKI-on (**Figure 4D,E**). Thus, increasing the local concentration of MLN8237 enhances drug action by approximately 2-fold at this subcellular location. These data demonstrate the versatility of the LoKI system as a pharmacological platform to block distinct kinases that operate at mitotic centrosomes.

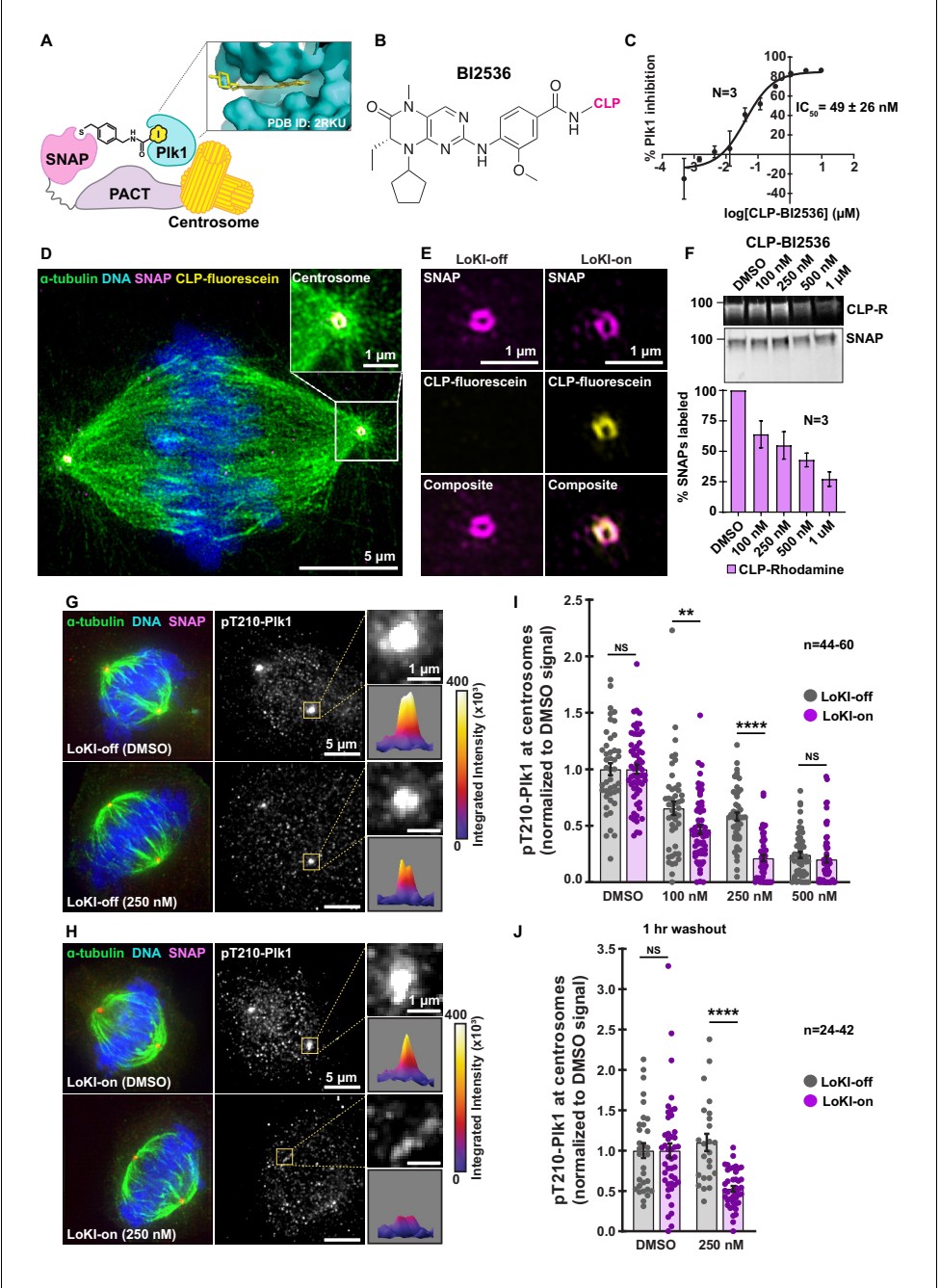

**Figure 2.** Validation of the LoKI platform. (A) Schematic of a centrosome-directed LoKI platform. SNAP-PACT fusion proteins conjugate CLP-linked Plk1 inhibitors at centrosomes. Inset depicts BI2536 in the ATP-binding pocket of Plk1. (B) Chemical structure of CLP-BI2536. (C) Dose-response curve of in vitro Plk1 inhibition with CLP-BI2536. (D) Structured illumination microscopy (SIM) of a LoKI-on U2OS cell labeled with CLP-fluorescein. Immunofluorescent detection of α-tubulin (green), DNA (blue), mCherry-SNAP-PACT (magenta) and CLP-fluorescein (yellow). Magnification of SNAP and CLP-fluorescein co-distribution at a centrosome (inset). (E) SIM micrographs of LoKI-off (left) and LoKI-on (right) U2OS cells. SNAP expression (top, magenta), CLP-fluorescein conjugation (mid, yellow) and composite images (bottom) are depicted. (F) Pulse-chase experiments measuring CLP-BI2536's ability to block CLP-rhodamine conjugation to LoKI-on. In-gel rhodamine fluorescence (top), immunoblot of SNAP loading controls (mid), and fluorescence quantification of pulse-chase experiments (bottom). (G, H) Immunofluorescence of representative mitotic LoKI-off (G) and LoKI-on (H) U2OS cells treated with DMSO or 250 nM CLP-BI2536 for 4 hr. Composite images (left) show α-tubulin (green), DNA (blue), and SNAP (magenta). Immunofluorescent detection of pT210-Plk1 (mid, gray) as an index of kinase activity. 5X magnification of centrosomal pT210-Plk1 signals and surface plots measuring integrated intensity of pT210-Plk1 signal (insets). (I, J) Quantification of centrosomal pT210-Plk1 immunofluorescence for LoKI-expressing cells. Points represent individual cells (n). Data normalized to DMSO. Application of DMSO or CLP-BI2536 for 4 hr, (I) 100 nM, LoKI-off, n = 46, LoKI-on, n = 59, **p=0.0059; 250 nM, LoKI-off, n = 46, LoKI-on, n = 46, ****p<0.0001 and drug treatment followed by 1 hr washout (J) 250 nM,

*Figure 2 continued*

LoKI-off, *n* = 24, LoKI-on, *n* = 42, ****p<0.0001. Experiments were conducted at least three times (N = 3) and *P* values were calculated by unpaired two-tailed Student's t-test. Data are mean ± s.e.m. NS, not significant. Source files for analysis of pulse-chase experiments are available in *Figure 2—source data 1* and for quantification of pT210-Plk1 are available in *Figure 2—source data 2*.

The online version of this article includes the following video, source data, and figure supplement(s) for figure 2:

**Source data 1.** Analysis for pulse-chase experiments with CLP-BI2536 in SNAP-PACT cells.
**Source data 2.** Raw analysis for pT210-Plk1 signal.
**Figure supplement 1.** Validation of the LoKI system.
**Figure supplement 2.** Conjugation of CLP-BI2536 to LoKI-on.
**Figure supplement 2—source data 1.** Analysis for pulse-chase time course experiments with CLP-BI2536 in SNAP-PACT cells.
**Figure supplement 3.** Characterization of Plk1 inhibition with CLP-BI2536.
**Figure supplement 4.** Non-normalized quantification of pT210-Plk1 signal at centrosomes.
**Figure 2—video 1.** CLP-substrates bind SNAP in LoKI-on but not LoKI-off cells.
https://elifesciences.org/articles/52220#fig2video1
**Figure 2—video 2.** CLP-substrates bind SNAP at centrosomes in LoKI-on cells.
https://elifesciences.org/articles/52220#fig2video2

## Combined Plk1 and AurA suppression at centrosomes more profoundly delays mitosis than global kinase inhibition

To investigate the coordinate activities of Plk1 and AurA at the centrosome (*Asteriti et al., 2015*) we took advantage of another feature of the LoKI-on platform, the ability to co-localize CLP-drug combinations via a dual SNAP conjugation moiety (*Figure 4F*). Live-cell imaging of U2OS cells expressing GFP-tagged histone 2B (*Figure 4G*, *Figure 4—figure supplement 2D*, *Figure 4—videos 1* and *2*) was used to calculate a baseline for mitotic timing (nuclear envelope breakdown to anaphase) as 35.1 min (*Figure 4H*). Mitosis was delayed by 19.4 min when CLP-BI2536 (250 nM) and CLP-MLN8237 (100 nM) were simultaneously applied to LoKI-off cells (*Figure 4G,H*). However, the same combination treatment prolonged mitosis 3-fold (59 min delay) when experiments were repeated in LoKI-on cells (*Figure 4G,H*). Moreover, the mitotic duration observed in LoKI-on cells after combination treatment extended beyond what was seen with either inhibitor alone (*Figure 4—figure supplement 3A,B*). These data accentuate the utility of LoKI-on as a means to direct drug combinations to defined cellular locations in space and time.

## Implementation of LoKI in live zebrafish embryos implicates Plk1 activity at centrosomes in coordinating mitoses during early development

Zebrafish provide an excellent model organism to test local drug action using the LoKI system because their transparency simplifies imaging analysis (*Zon and Peterson, 2005*). Zebrafish embryos were microinjected with mCherry-LoKI-on mRNA and allowed to develop for 5 hr until they reached ~50% epiboly (*Figure 5A,B*). Detection of mCherry fluorescence confirmed expression of the local drug-targeting construct (*Figure 5B*). Higher resolution imaging of fixed embryos confirmed accumulation of LoKI-on at centrosomes during interphase and mitosis (*Figure 5—figure supplement 1A*). Co-distribution of the SNAP moiety (magenta) with CLP-647 dye (yellow) confirmed assembly of the drug-targeting platform at centrosomes (*Figure 5—figure supplement 1B*). Microinjection of the Plk1 inhibitor adduct CLP-BI2536 (250 nM) permitted local drug delivery. Live-cell imaging 5 hr post injection exposed a range of adverse mitotic phenotypes. Mitotic spindles were visualized using a microtubule binding protein, EMTB-3xGFP (*Figure 5C*). Multipolar spindles, spindle orientation defects, and prolonged mitoses were evident in drug-treated embryos expressing LoKI-on (*Figure 5C*, *Figure 5—video 1*). Fixed-cell imaging of whole embryos revealed intact microtubule organization and few mitotic cells in LoKI-off embryos treated with CLP-BI2536 (*Figure 5D*). In contrast, drug-treated LoKI-on embryos exhibited microtubule abnormalities and a higher incidence of mitotic cells (*Figure 5E*). Fluorescent detection of the SNAP moiety confirmed centrosomal targeting of LoKI platforms (*Figure 5D,E*). Additional analyses correlated centrosomal inhibition of Plk1 with increased mitotic indices in LoKI-on embryos (*Figure 5F*). Conversely, control experiments in LoKI-off embryos showed that CLP-BI2536 had minimal effect, as indicated by a significantly higher proportion of interphase cells (*Figure 5F*). Thus, targeted delivery of kinase inhibitor drugs to

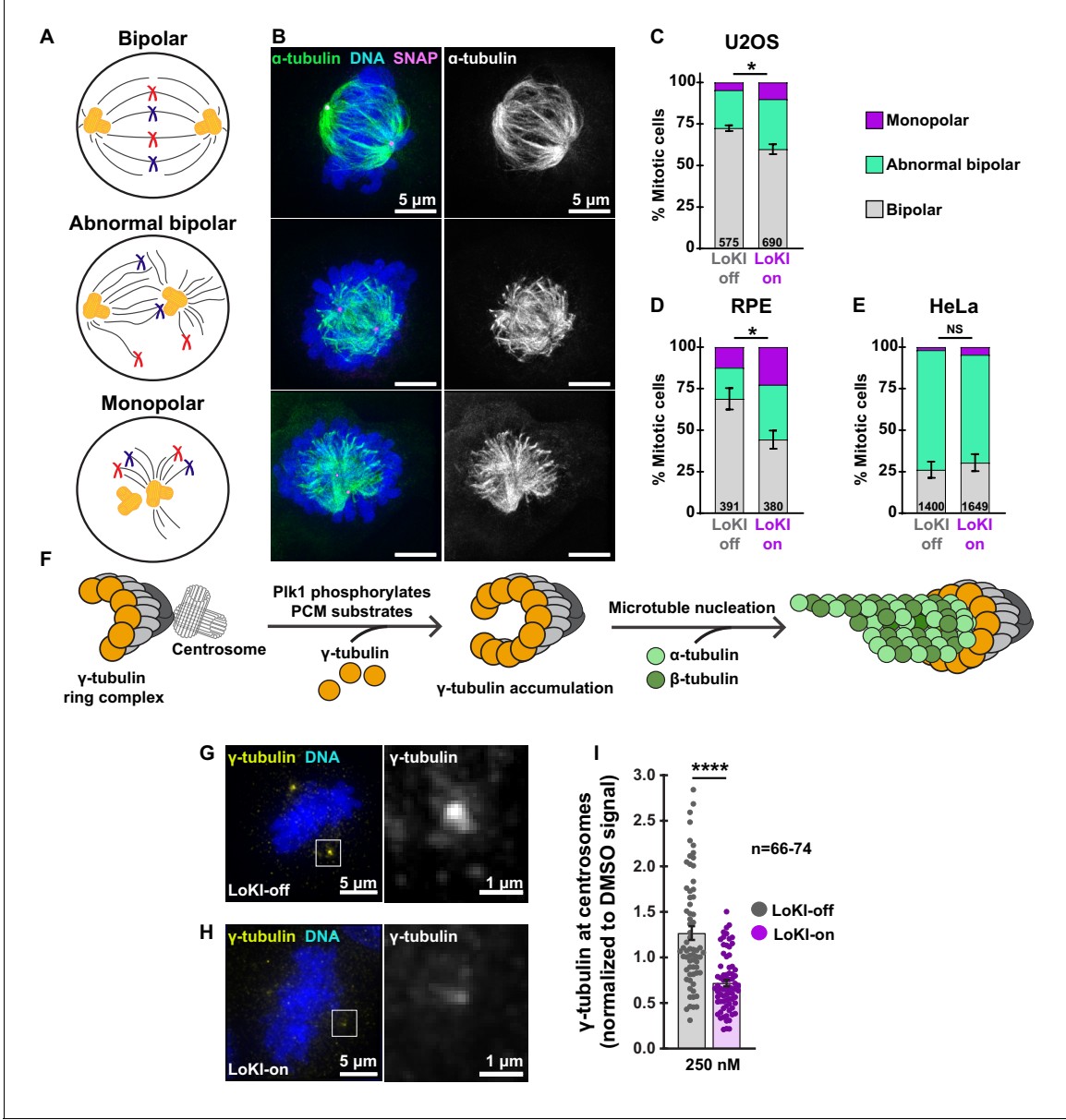

**Figure 3.** Centrosome-targeted Plk1 inhibitors perturb early mitotic events. (A) Schematic depicting bipolar (top), abnormal bipolar (mid), and monopolar (bottom) mitotic spindle classifications. (B) Representative composite (left) images show α-tubulin (green), DNA (blue), and SNAP (magenta) or α-tubulin (right, gray) staining alone for each spindle type in U2OS cells. (C–E) Spindle profile measurements of U2OS (C) RPE (D) and HeLa (E) cells treated with 250 nM CLP-BI2536 for 4 hr. Spindle profiling indicates the % of each spindle type in drug-treated LoKI-on and LoKI-off cells. Number of cells analyzed per condition are indicated; (C) n = 3, *p=0.0214; (D) n = 3, *p=0.0269. (F) Schematic depicting that Plk1 phosphorylation of pericentriolar substrates facilitates accumulation of γ-tubulin at centrosomes and microtubule nucleation. (G, H) Representative composite (left) images show γ-tubulin (yellow) and DNA (blue) in U2OS cells expressing LoKI-off (G) and LoKI-on (H) treated with 250 nM CLP-BI2536 for 4 hr, followed by a 1 hr washout. 5X magnified grayscale images of centrosomal γ-tubulin (right). (I) Quantification of centrosomal γ-tubulin immunofluorescence in LoKI-expressing cells. Points represent individual cells (n). Data normalized to DMSO. A ROUT (Q = 1%) outlier test was performed and two values were removed prior to performing statistical tests; LoKI-off, n = 66, LoKI-on, n = 74, ****p<0.0001. Experiments were conducted at least three times (N = 3) and *P* values were calculated by unpaired two-tailed Student's t-test. Data are mean ± s.e.m. NS, not significant. Source files for spindle profile analyses are available in *Figure 3—source data 1*.

The online version of this article includes the following source data and figure supplement(s) for figure 3:

**Source data 1.** Spindle profile analyses.
**Figure supplement 1.** Baseline controls evaluating mitotic defects.

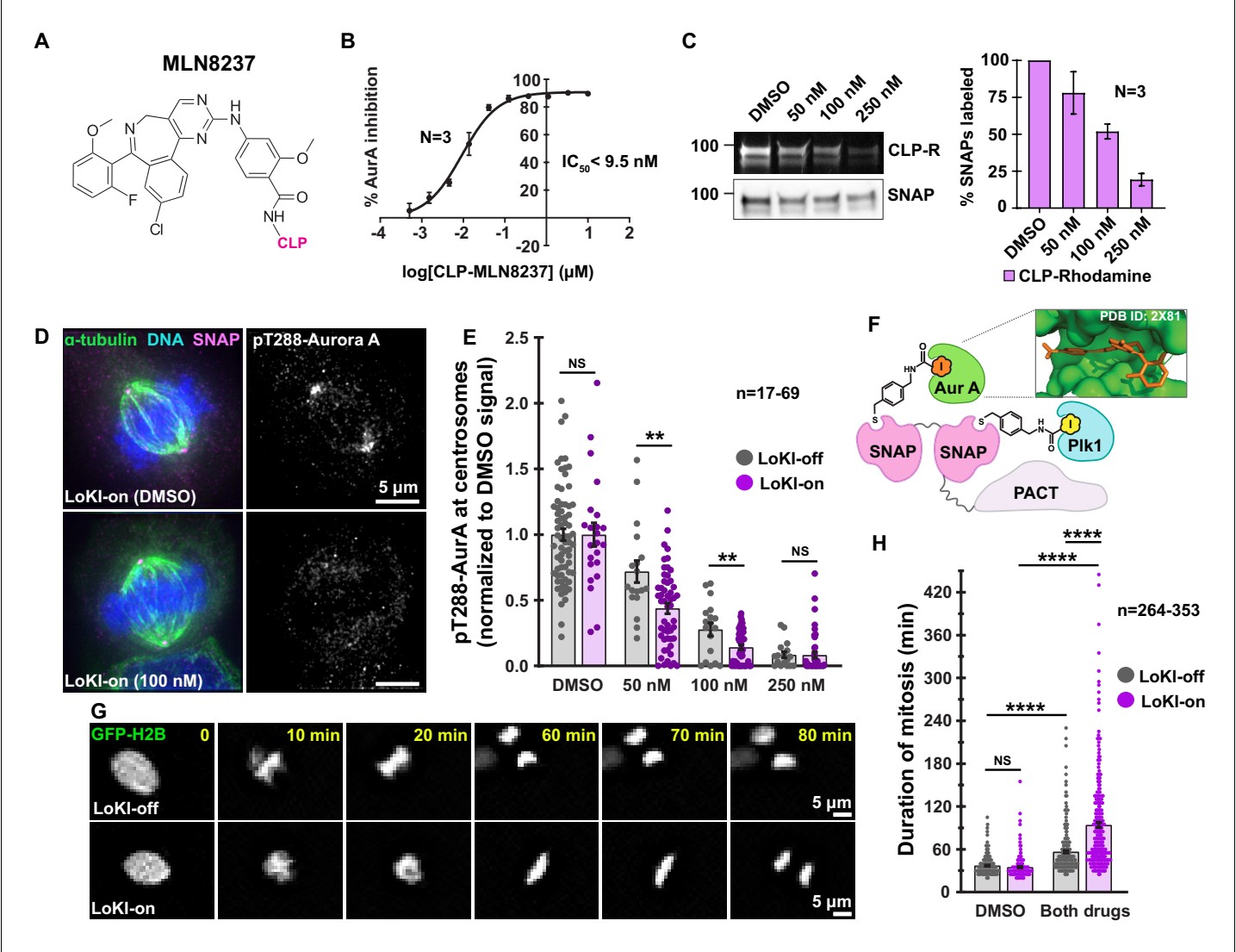

**Figure 4.** Combined Plk1 and AurA suppression at centrosomes more profoundly delays mitosis than global kinase inhibition. (**A**) Chemical structure of AurA kinase inhibitor CLP-MLN8237. (**B**) Dose-response curve of in vitro AurA kinase inhibition with CLP-MLN8237. (**C**) Pulse-chase experiments measuring CLP-MLN8237's ability to block CLP-rhodamine conjugation to LoKI-on. In-gel rhodamine fluorescence (top), immunoblot of SNAP loading controls (mid), and fluorescence quantification of pulse-chase experiments (bottom). (**D**) Immunofluorescence of representative mitotic LoKI-on U2OS cells treated with DMSO or 100 nM CLP-MLN8237 for 4 hr. Composite images (left) show α-tubulin (green), DNA (blue), and SNAP (magenta). Immunofluorescent detection of pT288-AurA (right, gray) as an index of kinase activity is depicted. (**E**) Quantification of centrosomal pT288-AurA immunofluorescence after 4 hr CLP-MLN8237 treatment; 50 nM, LoKI-off, $n = 18$, LoKI-on, $n = 53$, **$p=0.0014$; 100 nM, LoKI-off, $n = 18$, LoKI-on, $n = 48$, **$p=0.0026$. (**F**) Schematic of dual LoKI-on moiety conjugated to CLP-BI2536 and CLP-MLN8237. Inset depicts MLN8237 in the ATP-binding pocket of AurA. (**G**) Representative time-lapse images of mitotic LoKI-off (top) and LoKI-on (bottom) cells transiently expressing GFP-H2B. Cells were treated with a combination (both drugs) of 250 nM CLP-BI2536 and 100 nM CLP-MLN8237 and monitored for 18 hr. (**H**) Quantification of time-lapse experiments depicts duration of mitosis from nuclear envelope breakdown to the onset of anaphase; DMSO, LoKI-off, $n = 264$, both drugs, LoKI-off $n = 353$, ****$p<0.0001$; DMSO, LoKI-on, $n = 292$, both drugs, LoKI-on $n = 331$, ****$p<0.0001$; both drugs, LoKI-off, $n = 353$, both drugs, LoKI-on $n = 331$, ****$p<0.0001$. Points in (**E**) and (**H**) represent individual cells (n). Points in (**E**) are normalized to DMSO. Experiments were conducted at least three times (N = 3) and P values were calculated by unpaired two-tailed Student's t-test. Data are mean ± s.e.m. NS, not significant. Source files for analysis of pulse-chase experiments are available in *Figure 4—source data 1*.

The online version of this article includes the following video, source data, and figure supplement(s) for figure 4:

**Source data 1.** Analysis for pulse-chase experiments with CLP-MLN8237 in SNAP-PACT cells.
**Figure supplement 1.** CLP-MLN8237 and AurA activity assay.
**Figure supplement 2.** Characterization of AurA inhibition with CLP-MLN8237.
**Figure supplement 3.** Mitotic duration is prolonged in cells treated with centrosome-targeted Plk1 or AurA inhibitors.
**Figure 4—video 1.** Comparison of mitotic progression in DMSO-treated LoKI cells.

*Figure 4 continued on next page*

mitotic centrosomes induces a range of adverse mitotic phenotypes in developing embryos relative to global drug application.

## Targeting CLP-MLN8237 to kinetochores reveals that AurA-mediated Hec1 phosphorylation is a local event

Kinetochores are proteinaceous structures that ensure the proper attachment of spindle microtubules to the centromeric region of condensed chromatin (*Hinshaw and Harrison, 2018*) (*Figure 6A*). To further demonstrate the versatility of the LoKI system, we utilized a targeting domain from the kinetochore protein Mis12 (*Goshima et al., 2003*) (*Figure 6A*). Inducible expression of the mCherry-tagged fusion was accomplished by a doxycycline-inducible promoter (*Figure 6B*, *Figure 6—figure supplement 1A*). Immunoblot detection of Mis12-LoKI-on persisted up to 4 hr upon removal of doxycycline (*Figure 6—figure supplement 1B*). Immunofluorescent staining revealed that mCherry-tagged Mis12-LoKI-on co-localized with centromeric DNA (anti-centromere antibodies (ACA), cyan) at kinetochores during mitosis (*Figure 6C*, *Figure 6—video 1*). Counterstaining with α-tubulin antibodies (green) revealed the mitotic spindle (*Figure 6C*). SIM imaging revealed that a CLP-dye (CLP-647, yellow) accumulated with the SNAP moiety (magenta) at kinetochores of Mis12-LoKI-on cells (*Figure 6D*). In contrast, recruitment of CLP-647 was not evident in Mis12-LoKI-off cells (*Figure 6D*). Line plot analyses of the CLP-647 signal in selected kinetochores emphasizes this result (*Figure 6E*). Pulse-chase experiments established that incubation with 100–250 nM of the AurA inhibitor adduct CLP-MLN8237 for 4 hr at 37°C was the optimal drug regimen (~50% of drug binding sites occupied; *Figure 6F*). Parallel, validation studies were performed with CLP-BI2536 (*Figure 6—figure supplement 1C*).

Roles for AurA at centrosomes and mitotic spindles are well documented (*Nikonova et al., 2013*). However, recent reports have implicated AurA as a modulator of microtubule attachment to kinetochores (*Chmátal et al., 2015*; *Ye et al., 2015*). At this location AurA phosphorylates serine 69 in Hec1, a subunit of the NDC80 complex (*DeLuca et al., 2018*). This local phosphorylation stabilizes microtubule-kinetochore interaction to safeguard chromosome segregation (*DeLuca et al., 2018*). However, the proximity of centrosomes to kinetochores in early mitosis has hampered attempts to resolve the contribution of discrete AurA pools. Therefore, immunofluorescent detection of pS69-Hec1 served as an index for local AurA kinase activity at kinetochores (*Figure 6G,H*). As before, counterstaining for α-tubulin (green) and DNA (blue) revealed the mitotic spindle (*Figure 6G,H*). In cells expressing Mis12-LoKI-off, incubation with 100 nM CLP-MLN8237 caused a negligible decrease in pS69-Hec1 signal as compared to DMSO-treated controls (*Figure 6G,I*). Conversely, in Mis12-LoKI-on cells the pS69-Hec1 signal was reduced to 59.8% (*Figure 6H,I*). Representative heat maps further illustrate this phenomenon (*Figure 6G,H*). Importantly, the pS69-Hec1 signal at centrosomes was unaffected by drug treatments (*Figure 6G,H*, *Figure 6—figure supplement 2A*). Amalgamated data from three independent experiments reveal that this trend persisted even at higher concentrations (*Figure 6I*). Immunoblot analyses of nocodazole-synchronized cells collected via mitotic shake-off further support our findings (*Figure 6—figure supplement 2B*). Finally, when SNAP-PACT LoKI-expressing cells were used to sequester CLP-MLN8237 at centrosomes, we no longer observed a reduction in pS69-Hec1 signal at kinetochores (*Figure 6—figure supplement 2C*). These data further support a role for AurA at the kinetochore. Ultimately, our findings illustrate how LoKI platforms can be adapted to pharmacologically investigate kinase signaling at distinct subcellular locations within the mitotic cell.

## Discussion

Cells have evolved a highly organized architecture that is segregated into functionally distinct microenvironments (*Figure 6J*). However, traditional methods of drug delivery do not account for this

exquisite degree of molecular organization. Conventional approaches flood cells with bioactive compounds, masking the unique contributions of individual kinases at distinct subcellular locations. Although it is well established that Plk1 and AurA coordinate various aspects of cell division, current drug-targeting strategies limit our ability to decode the spatiotemporal regulation of these events (*Barr et al., 2004*; *Combes et al., 2017*; *Lens et al., 2010*). Studying molecular scaffolds that form complexes with these key mitotic enzymes provides important mechanistic insight into how these processes are coordinated (*Figure 1*; *Hehnly et al., 2015*). Moreover, designing pharmacological tools that restrict the spatial and temporal action of kinase inhibitor drugs is paramount to deciphering local kinase action. In this study, we discovered that the anchoring protein Gravin is required for organizing active pools of Plk1 and AurA at centrosomes (*Figure 1C,D*). These data support previous findings in which Gravin loss led to increased Plk1 mobility and aberrant CEP215 phosphorylation (*Canton et al., 2012*; *Hehnly et al., 2015*; *Colicino et al., 2018*). Thus, we suggest that Gravin constrains enzymes in a signaling island to provide spatiotemporal control of kinase activity. Conversely, depletion of this anchoring protein alters Gravin-Plk1 and Gravin-AurA protein-protein interactions which underlie healthy cellular function. This further emphasizes a need for designing strategies that inhibit kinase activity locally. For this reason, we developed a novel chemical-biology tool, LoKI, to more precisely probe the actions of Plk1 and AurA at centrosomes and kinetochores. Previous work from Gower and colleagues utilized antibody mimetics to target promiscuous inhibitor drugs to specific kinases (*Gower et al., 2016*). Our strategy advances this technology by combining AKAP-targeting domains with SNAP-tagging technologies. Additionally, we direct selective ATP-competitive inhibitors to specific subcellular locations to achieve local kinase inhibition. By combining biochemical approaches, quantitative imaging, and live-cell microscopy we reveal that local targeting of Plk1 and AurA kinase inhibitor drugs disrupts substrate phosphorylation, spindle organization, and mitotic duration more profoundly than global drug distribution. Thus organellar targeting of drugs offers a new means to advance the investigation of broad-spectrum kinases at precise locations.

Previous studies suggest that Plk1 phosphorylates pericentriolar substrates that coordinate γ-tubulin accumulation at mitotic centrosomes to facilitate microtubule nucleation (*Lane and Nigg, 1996*; *Haren et al., 2009*; *Xu and Dai, 2011*). We advance this concept and extend these findings by demonstrating that centrosomal inhibition of Plk1 prevents accumulation of γ-tubulin and correct organization of bipolar mitotic spindles (*Figure 3*). Thus, by using the LoKI system we are able to definitively establish that Plk1 activity at mitotic centrosomes is a driver in these processes. Furthermore, the utility of LoKI drug targeting was underscored by our in vivo studies using zebrafish embryos. We provide evidence that embryos treated with centrosome-targeted Plk1 inhibitors have more microtubule abnormalities than those treated with a non-localized inhibitor (*Figure 5C–E*). These data implicate centrosome-localized pools of Plk1 in coordinating mitotic events such as spindle organization and mitotic progression during early zebrafish development. In a broader context, we show that local targeting of Plk1 inhibitors in developing organisms offers an innovative precision technique to probe local drug action.

Another key advance in our studies came with the discovery that AurA-mediated Hec1 phosphorylation is a spatially-coordinated event that occurs at kinetochores. We provide quantitative imaging (*Figure 6G–I*, *Figure 6—figure supplement 2C*) and biochemical (*Figure 6—figure supplement 2B*) data that implicates AurA activity at this mitotic substructure. Although this kinase was originally thought to reside exclusively at centrosomes and mitotic spindles, our findings extend recent evidence for the existence of an AurA pool at kinetochores (*DeLuca, 2017*). We reveal that kinetochore-targeted MLN8237 reduces pS69-Hec1 signal more drastically than globally distributed drug (*Figure 6H–I*). Our data suggest that even during prometaphase, when kinetochores may encounter centrosome-associated AurA, this phosphorylation is solely a kinetochore-associated event. Furthermore, when we target the AurA inhibitor to centrosomes and measure pS69-Hec1 at kinetochores we no longer see a loss of pS69-Hec1 signal (*Figure 6—figure supplement 2C*). Thus, we show that centrosome-associated AurA is not likely responsible for this phosphorylation event as has been previously suggested (*Chmátal et al., 2015*; *Ye et al., 2015*). More importantly, these findings uncover that S69-Hec1 phosphorylation is a local event that depends on AurA activity at kinetochores. This allows us to postulate that isolated pockets of AurA may act independently and concurrently to orchestrate complex cellular events.

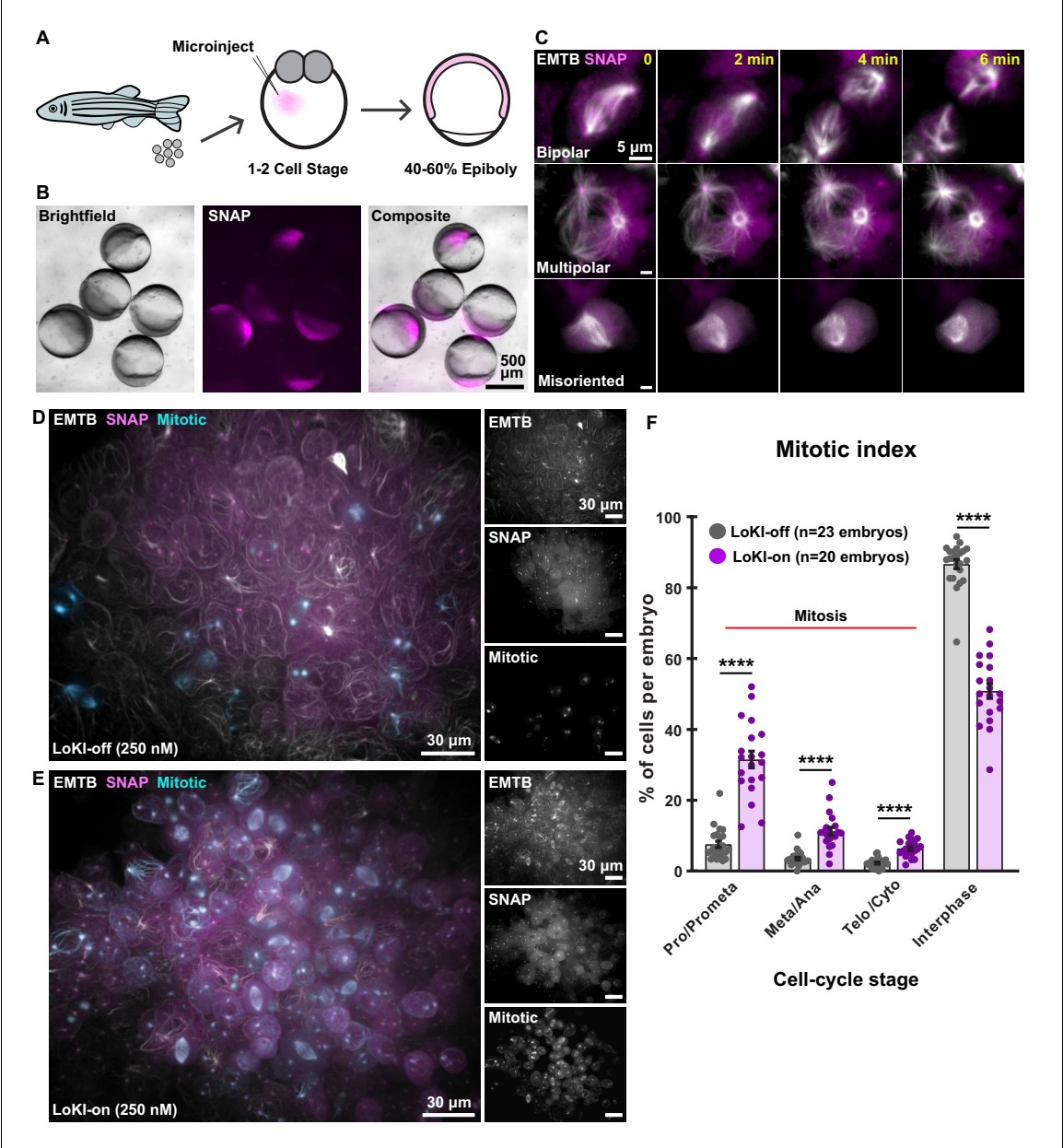

**Figure 5.** In vivo implementation of LoKI implicates Plk1 activity at centrosomes in coordinating mitoses during early development. (A) Schematic of experimental scheme. Microinjection of LoKI mRNAs into zebrafish embryos occurs at the 1–2 cell stage. Live-cell imaging is conducted at ~50% epiboly. (B) Zebrafish embryos (left, brightfield) depicting regional expression of SNAP (mid, magenta) at ~50% epiboly. Composite images (right) depict expression of SNAP only in the cells of the zebrafish embryos. (C) Centrosomal delivery of Plk1 inhibitors perturb cell division in zebrafish embryos. Time-lapse images of dividing cells embedded in LoKI-on zebrafish embryos 5 hr post application of 250 nM CLP-BI2536. Representative examples of normal bipolar spindles (top) multipolar spindles (mid) and spindle orientation defects (bottom) are presented. Composite images show microtubule marker EMTB-3xGFP (white) and SNAP (magenta). (D, E) 3D-rendered images depict incidence of mitotic cells and general organization of whole LoKI-off (D) or LoKI-on (E) zebrafish embryos treated with 250 nM CLP-BI2536. EMTB (white and top inset), SNAP (magenta and mid inset), and mitotic cells (cyan and bottom inset) are shown. (F) Graph depicting mitotic index measurements for LoKI-expressing embryos; LoKI-off, n = 23, LoKI-on, n = 20, ****p<0.0001. Each point represents % of cells per individual embryo (n). Experiments were conducted at least three times (N = 3) and P values were calculated by unpaired two-tailed Student's t-test. Data are mean ± s.e.m.

The online version of this article includes the following video and figure supplement(s) for figure 5:

**Figure supplement 1.** Validation of LoKI expression in zebrafish embryos.
**Figure 5—video 1.** Mitotic defects in drug-treated LoKI-on embryos.
https://elifesciences.org/articles/52220#fig5video1

The versatility of this new chemical-biology platform is demonstrated in three ways. First, this approach works in a variety of cell types and microinjection of LoKI mRNA into live zebrafish embryos permits local drug targeting in vivo (*Figure 5*). We foresee that LoKI platforms will be adapted to acutely probe local signaling in other genetically tractable organisms. Second, while derivatized Plk1 and AurA drugs delineate roles for each mitotic kinase, conjugation of chloropyrimidine (CLP) to other ATP analogs offers a general method to synthesize localizable inhibitors for additional members of the kinome (*Gower et al., 2016*). However, it is worth noting that the reduced cell permeability of certain CLP-drug conjugates, including CLP-BI2536, may necessitate their use at approximately 10-fold higher concentrations than the unmodified drugs (*Figure 2I* versus *Figure 2—figure supplement 3B*, *Figure 4E* versus *Figure 4—figure supplement 2B*). Additionally, derivatization of certain inhibitors may sterically hinder their access to the ATP-binding pockets of some kinases or, as is the case of the PKA antagonist H89, the lack of a functional group prevents CLP derivatization. Third, plasma membrane and mitochondrial targeting domains from AKAP79 and dAKAP1 expand the repertoire of subcellular compartments reached by LoKI platforms (*Figure 6J*).

Although our strategy uncovers local contributions of anchored kinase pools within the cell, certain limitations to our current approach exist. For example, we treat cells for 4 hr with CLP-conjugated drug adducts to achieve sufficient inhibitor targeting (~50% of drug binding sites occupied; *Figure 2F*, *Figure 6F*). This is a relatively long time period in the context of measuring cell-cycle events. Likewise, in cells we find that high concentrations of CLP-drug (100 nM) are required to produce equivalent effects as 10 nM of non-derivatized MLN8237 (*Figure 4E* versus *Figure 4—figure supplement 2B*). We postulate that reduced cell permeability of certain CLP-drug conjugates may account for both of the aforementioned findings. As a result, this necessitates long incubation periods and application of higher concentrations of drug. Finally, it is possible that kinase inhibitor drugs directed to centrosomes have off-target effects at nearby structures such as spindle microtubules. We envision that future advancements of this platform would include photo-caged inhibitors that are inert until they are ready to be released at the site of desired inhibition (*Ellis-Davies, 2007*). Employing this strategy would more strictly define the range of inhibitor action, provide another level of control to the LoKI system, and allow us to better delineate the effects of global versus local inhibition.

One exciting feature of our LoKI platform is the ability to co-localize CLP-drug combinations via a dual SNAP conjugation moiety (*Figure 4F*). Although in our study this allowed combined inhibition of Plk1 and AurA at centrosomes, we hope that future work will advance on our strategy and provide a system that utilizes multiple self-labeling enzymes to deliver distinct inhibitors to the same location. Employing orthogonal tagging systems such as CLIP-tag or Halo-tag in tandem with SNAP-tag could eliminate the possibility that CLP-BI2536 and CLP-MLN8237 compete for conjugation to the same targeting moiety. Nonetheless, by exploiting our knowledge of how AKAPs compartmentalize signaling enzymes we have developed tools that define the local kinase terrain at the angstrom level. This will allow investigators to probe local signaling events at a level of precision that has not been possible before.

# Materials and methods

**Key resources table**

| Reagent type (species) or resource | Designation | Source or reference | Identifiers | Additional information |
|---|---|---|---|---|
| Antibody | Alpha-tubulin, clone DM1A | Sigma Aldrich | T9026 Mouse monoclonal RRID:AB_477593 | IF (1:500) |
| Antibody | Alpha-tubulin-FITC, clone DM1A | Sigma Aldrich | F2168 Mouse monoclonal RRID:AB_477593 | IF (1:200) |

*Continued on next page*

*Continued*

| Reagent type (species) or resource | Designation | Source or reference | Identifiers | Additional information |
|---|---|---|---|---|
| Antibody | Amersham ECL Mouse IgG, HRP-linked F(ab')$_2$ fragment (from sheep) | GE Life Sciences | NA9310 | WB (1:10000) |
| Antibody | Amersham ECL Rabbit IgG, HRP-linked F(ab')$_2$ fragment (from donkey) | GE Life Sciences | NA9340 | WB (1:10000) |
| Antibody | Aurora A | Sigma Aldrich | SAB2500135 Goat polyclonal | WB (1:1000) |
| Antibody | Centromere (ACA) | Antibodies Inc. | 15-234-0001 Human polyclonal | IF (1:200) |
| Antibody | Phospho-Aurora A (T288), clone C39D8 | Cell Signaling | 3079 Rabbit polyclonal | IF (1:500); WB (1:1000) |
| Antibody | Donkey anti-goat IgG-HRP | Santa Cruz | sc-2020 | WB (1:10000) |
| Antibody | Donkey anti-Mouse IgG, Alexa Fluor 488 | Invitrogen | A-21202 | IF (1:500) |
| Antibody | Donkey anti-Mouse IgG, Alexa Fluor 555 | Invitrogen | A-31570 | IF (1:500) |
| Antibody | Donkey anti-Mouse IgG, Alexa Fluor 647 | Invitrogen | A-11126 | IF (1:500) |
| Antibody | Donkey anti-Rabbit IgG, Alexa Fluor 488 | Invitrogen | A-21206 | IF (1:500) |
| Antibody | Donkey anti-Rabbit IgG, Alexa Fluor 555 | Invitrogen | A-31572 | IF (1:500) |
| Antibody | Donkey anti-Rabbit IgG, Alexa Fluor 647 | Invitrogen | A-31573 | IF (1:500) |
| Antibody | DyLight 405 AffiniPure Donkey Anti-Human IgG (H+L) | Jackson Labs | RRID: AB_2340553 | IF (1:500) |
| Antibody | GAPDH-HRP | Novus | NB110-40405 Mouse monoclonal RRID: AB_669249 | WB (1:2000) |
| Antibody | Gamma-tubulin | Abcam | 11317 Rabbit polyclonal RRID: AB_297921 | IF (1:1500) |
| Antibody | Gravin, clone JP74 | Sigma Aldrich | G3795 Mouse monoclonal | WB (1:1000) |
| Antibody | Gravin, clone R3698 | (*Nauert et al., 1997*) | Rabbit polyclonal | IF (1:1000) |
| Antibody | Phospho-Gravin (T766) | (*Canton et al., 2012*) | Rabbit polyclonal | IF (1:1000) |
| Antibody | Phospho-Hec1 (S69) | (*DeLuca et al., 2018*) | Rabbit polyclonal | IF (1:3000) WB (1:3000) |
| Antibody | Phospho-Plk1 (T210) | Biolegend | 628901 Rabbit polycolonal RRID: AB_439786 | IF (1:500); WB (1:1000) |

*Continued on next page*

*Continued*

| Reagent type (species) or resource | Designation | Source or reference | Identifiers | Additional information |
|---|---|---|---|---|
| Antibody | Plk1, clone 35–206 | Millipore | 05–844 Mouse monoclonal RRID:AB_11213632 | IF (1:500); WB (1:1000) |
| Antibody | SNAP-tag | New England Biolabs | P9310S Rabbit polyclonal | WB (1:1000) |
| Cell line (*H. sapein*) | HEK293 Control shRNA stable cell line | (*Canton et al., 2012*) | | Maintained in Scott lab in DMEM supplemented with 10% FBS under 4 ug/mL Puromycin selection |
| Cell line (*H. sapein*) | HEK293 Gravin shRNA stable cell line | (*Canton et al., 2012*) | | Maintained in Scott lab in DMEM supplemented with 10% FBS under under 4 ug/mL Puromycin selection |
| Cell line (*H. sapein*) | HeLa | From L. Wordeman lab, origin ATCC | | Maintained in the Scott Lab in DMEM supplemented with 10% FBS |
| Cell line (*H. sapein*) | hTERT-RPE | Gift from P. Jallepalli lab, origin ATCC | | Maintained in the Scott Lab in DMEM/F-12, Hepes with 10% FBS |
| Cell line (*M. musculus*) | MEF Gravin wildtype | (*Hehnly et al., 2015*) | | |
| Cell line (*M. musculus*) | MEF Gravin knockout | (*Hehnly et al., 2015*) | | |
| Cell line (*H. sapein*) | U2OS | ATCC | HTB-96 | Maintained in the Scott Lab in DMEM supplemented with 10% FBS |
| Chemical compound, drug | Alisertib, MLN8237 | AdooQ | A10004-10nM-D | Manufacturer's instructions |
| Chemical compound, drug | BI2536 | AdooQ | A10134-50 | Manufacturer's instructions |
| Chemical compound, drug | Beta-mercaptoethanol (BME) | Sigma Aldrich | M6250 | |
| Chemical compound, drug | CLP-BI2536 | This paper | | See 'synthesis of CLP-reagents' |
| Chemical compound, drug | CLP-MLN8237 | This paper | | See 'synthesis of CLP-reagents' |
| Chemical compound, drug | CLP-rhodamine | This paper | | See 'synthesis of CLP-reagents' |
| Chemical compound, drug | DAPI | Thermo Fisher | 62248 | IF (1:1000) |
| Chemical compound, drug | Dimethylsulfoxide (DMSO) | Pierce | TS-20688 | Manufacturer's instructions |
| Chemical compound, drug | DMEM FluoroBrite | Life Technologies | A1896701 | |

*Continued on next page*

*Continued*

| Reagent type (species) or resource | Designation | Source or reference | Identifiers | Additional information |
|---|---|---|---|---|
| Chemical compound, drug | DMEM/F-12 Hepes | Life Technologies | 11330057 | |
| Chemical compound, drug | DMEM, high glucose | Life Technologies | 11965118 | |
| Chemical compound, drug | Doxycycline hyclate | Sigma Aldrich | 24390-14-5 | |
| Chemical compound, drug | Fetal Bovine Serum | Thermo Fisher | A3382001 | |
| Chemical compound, drug | Lipofectamine 2000 Transfection Reagent | Invitrogen | 11668027 | |
| Chemical compound, drug | NuPAGE LDS Sample Buffer 4X | Thermo Fisher | NP0008 | |
| Chemical compound, drug | Opti-MEM I Reduced Serum Medium, no phenol red | Life Technologies | 11058021 | |
| Chemical compound, drug | ProLong Diamond Antifade Mountant | Life Technologies | P36961 | Manufacturer's instructions |
| Chemical compound, drug | Polybrene | Santa Cruz | 134220 | Manufacturer's instructions |
| Chemical compound, drug | Puromycin dihydrochloride | Santa Cruz | 58-58-2 | 4 ug/mL |
| Chemical compound, drug | SNAP-Cell Fluorescein | New England Biolabs | S9107S | Manufacturer's instructions |
| Chemical compound, drug | SNAP-Cell 647-SiR | New England Biolabs | S9102S | Manufacturer's instructions |
| Chemical compound, drug | SuperSignal West Dura Extended Duration Substrate | Thermo Fisher | 34075 | |
| Chemical compound, drug | TransIT-LT1 Transfection Reagent | Mirus | MIR2300 | |
| Chemical compound, drug | Trypsin-EDTA (0.25%), phenol red | Gibco | 25200056 | |
| Commercial assay or kit | BCA Protein Assay Kit | Thermo Fisher | 23227 | |
| Commercial assay or kit | QuikChange II XL kit | Aligent | 200522 | |
| Commercial assay or kit | GeneJET Genomic DNA purification kit | Thermo Fisher | K0721 | |
| Peptide, recombinant protein | Aurora A, active | Invitrogen | PV3612 | |
| Peptide, recombinant protein | Plk1, active | SignalChem | P41-10H | |

*Continued on next page*

*Continued*

| Reagent type (species) or resource | Designation | Source or reference | Identifiers | Additional information |
|---|---|---|---|---|
| Recombinant DNA reagent | EMTB-3XGFP | Addgene | | pCS2+ backbone |
| Recombinant DNA reagent | GFP-H2B | Addgene | | pEGFP-N1 backbone |
| Recombinant DNA reagent | His6-SNAP-tag | Addgene | | pMCSG7 backbone |
| Recombinant DNA reagent | pMD2.G | | RRID: Addgene_12259 | gift from Didier Trono; Addgene plasmid #12259 |
| Recombinant DNA reagent | psPAX2 | | RRID: Addgene_12260 | gift from Didier Trono; Addgene plasmid #12259 |
| Recombinant DNA reagent | SNAP-PACT | This paper | | In-house modified pLIX402 backbone (gift from David Root; Addgene plasmid #41394) |
| Recombinant DNA reagent | SNAP-PACT (C144A) | This paper | | In-house modified pLIX402 backbone (gift from David Root; Addgene plasmid #41394) |
| Recombinant DNA reagent | SNAP-Mis12 | This paper | | In-house modified pLIX402 backbone (gift from David Root; Addgene plasmid #41394) |
| Recombinant DNA reagent | SNAP-Mis12 (C144A) | This paper | | In-house modified pLIX402 backbone (gift from David Root; Addgene plasmid #41394) |
| Recombinant DNA reagent | SNAP-AKAP79 | This paper | | In-house modified pcDNA3.1+ backbone (Life Technologies) |
| Recombinant DNA reagent | SNAP-dAKAP1 | This paper | | In-house modified pcDNA3.1+ backbone (Life Technologies) |
| Software, algorithm | Fiji/ImageJ | ImageJ (http://imagej.nih.gov/ij/) | | |
| Software, algorithm | GraphPad Prism | GraphPad Prism (https://graphpad.com) | | |
| Software, algorithm | Imaris | Bitplane | | |
| Software, algorithm | SoftWoRx | GE Healthcare | | |
| Other | 1.5 poly-D-lysine coated coverslips | neuVitro | GG-12–1.5-pdl | |
| Other | AnykD Criterion TGX Precast Midi Protein Gel | Biorad | 5671124 | |
| Other | Bolt 4–12% Bis-Tris Plus Gels | Invitrogen | NW04120BOX | |
| Other | Scienceware cloning discs | Sigma Aldrich | Z374431 | |
| Other | μ-Slide 4 Well Glass Bottom: # 1.5H (170 μm + /- 5 μm) D 263 M Schott glass | (Ibidi)Pierce | 80426 | |

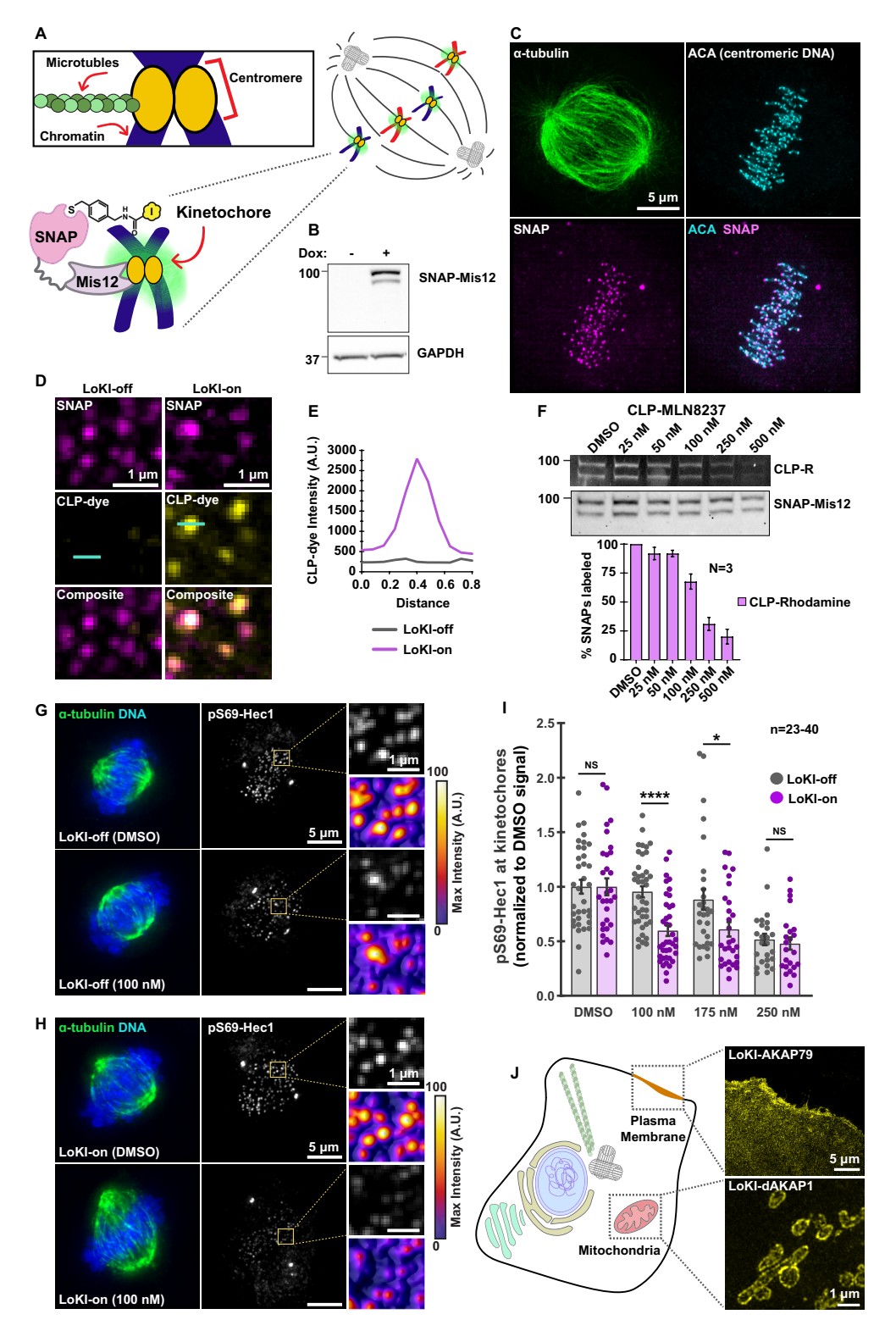

**Figure 6.** Kinetochore-targeted CLP-MLN8237 reveals that AurA-mediated Hec1 phosphorylation is a local event. (**A**) Schematic of microtubule association with centromeric chromatin at kinetochores. Diagram of LoKI-Mis12 securing CLP-linked inhibitors to kinetochores (inset). (**B**) Immunoblot confirming SNAP-Mis12 (top) expression after induction with doxycycline for 72 hr and GAPDH loading controls (bottom). (**C**) Representative SIM micrograph depicting α-tubulin (green), mCherry-SNAP-Mis12 (magenta) and centromeric DNA marker ACA (cyan) in U2OS cells. Composite image
*Figure 6 continued on next page*

Figure 6 continued

depicts co-distribution of LoKI-Mis12 with ACA. (D) SIM micrographs of LoKI-off (left) and LoKI-on (right) U2OS cells. SNAP expression (top, magenta), CLP-647 conjugation (mid, yellow) and composite images (bottom) are depicted. Line on CLP-dye was used to generate panel E plot. (E) Line plot of CLP-dye signal from a representative kinetochore in panel D. (F) Pulse-chase experiments measuring CLP-MLN8237's ability to block CLP-rhodamine conjugation to LoKI-on. In-gel rhodamine fluorescence (top), immunoblot of SNAP loading controls (mid), and fluorescence quantification of pulse-chase experiments (bottom). (G, H) Immunofluorescence of representative mitotic LoKI-off (G) and LoKI-on (H) U2OS cells treated with DMSO or 100 nM CLP-MLN8237 for 4 hr. Composite images (left) show α-tubulin (green) and DNA (blue). Immunofluorescence detection of pS69-Hec1 (mid and top right insets, gray). Heat maps (bottom right) depict maximum intensity measurements of pS69-Hec1 signal from representative insets. (I) Quantification of pS69-Hec1 immunofluorescence at kinetochores after 4 hr CLP-MLN8237 treatment; 100 nM, LoKI-off, $n = 40$, LoKI-on, $n = 39$, ****$p<0.0001$; 175 nM, LoKI-off, $n = 29$, LoKI-on, $n = 29$, *$p=0.0212$. Points represent individual cells (n). Data normalized to DMSO. Experiments were conducted at least three times ($N = 3$) and $P$ values were calculated by unpaired two-tailed Student's t-test. Data are mean ± s.e.m. NS, not significant. (J) Implementation of LoKI at other subcellular locations shows SNAP targeting to plasma membrane via AKAP79 and mitochondrial localization via d-AKAP1. Source files for analysis of pulse-chase experiments are available in *Figure 6—source data 1*.

The online version of this article includes the following video, source data, and figure supplement(s) for figure 6:

**Source data 1.** Analysis for pulse-chase experiments with CLP-MLN8237 in SNAP-Mis12 cells.
**Figure supplement 1.** Validation of Mis12-LoKI platforms.
**Figure supplement 1—source data 1.** Analysis for pulse-chase experiments with CLP-BI2536 in SNAP-Mis12 cells.
**Figure supplement 2.** Characterization of AurA inhibition with CLP-MLN8237.
**Figure 6—video 1.** CLP-substrates bind SNAP at kinetochores in Mis12-LoKI-on cells.
https://elifesciences.org/articles/52220#fig6video1

## Plasmid constructs

SNAP, mCherry, eGFP, PACT, Mis12, AKAP79, and dAKAP1 components and were individually PCR amplified with overlapping ends and/or Gateway 'att' sites and assembled using Gibson Cloning. Gateway cloning was carried out to subclone SNAP constructs into pLIX402 (a gift from David Root; Addgene plasmid #41394) for PACT and Mis12 studies or pcDNA3.1+ (Life Technologies) for AKAP79 and dAKAP1. To generate mutant SNAP, site-directed mutagenesis was performed with a QuikChange II XL kit (Aligent). GFP-H2B and EMTB-3xGFP constructs were used for live-cell imaging studies. Constructs were verified by Sanger sequencing.

## Cell culture and virus generation

Cells used to generate stable cell lines in this study originated as follows: U2OS (purchased from ATCC), HeLa (from L. Wordeman lab and maintained in-house), and hTERT-RPE (gift from P. Jalle-palli lab and maintained in-house). HeLa and hTERT-RPE cells were tested by STR at ATCC. Chang Liver cells, a HeLa contaminant, were detected in the HeLa line while hTERT-RPE cells were an exact match to ATCC cell line CRL-4000 (hTERT-RPE-1). U2OS, HeLa, and hTERT-RPE cells tested negative for mycoplasma contamination as assessed by the Universal Mycoplasma Detection Kit (ATCC 30-1012K). U2OS, HeLa, Control and Gravin shRNA HEK293 (*Canton et al., 2012*), and immortalized MEF (generated as described in *Hehnly et al. (2015)* and maintained in-house) cells were maintained in DMEM, high glucose and hTERT-RPE cells were maintained in DMEM/F-12, Hepes (Life Technologies) at 37°C and 5% CO2. All media was supplemented with 10% FBS. Infections for generation of stable SNAP cells were performed using lentiviral particles created in-house. In brief, SNAP pLIX402 vectors were transfected alongside pMD2.G and psPAX2 plasmids (gifts from Didier Trono; Addgene plasmid #12259 [RRID:Addgene_12259] and plasmid #12260 [RRID:Addgene_12260])) into HEK293 cells using Lipofectamine 2000 reagent (Invitrogen) in Opti-MEM (Life Technologies) media. Virus-containing supernatant was collected, passed through a. 45 μm filter, and transduced into cells in the presence of 1 μg/ul Polybrene (Santa Cruz). Cells were selected and maintained in supplemented media with 4 μg/mL Puromycin dihydrochloride (Santa Cruz). Single clones were isolated using Scienceware cloning discs (Sigma-Aldrich). Infections for generation of stable knockdown in HEK293 cells were performed with shRNA lentiviral particles (Santa Cruz Biotech) as described in *Canton et al. (2012)*. For expression of AKAP79, dAKAP1, and Gravin constructs in U2OS cells, transient transfections were performed using TransIT-LT1 reagent (Mirus) in Opti-MEM (Life Technologies) media according to manufacturer's instructions.

## Synthesis of CLP-reagents

### CLP-linker synthesis

**Chemical structure 1.** CLP-linker synthesis.

**1a** CLP-Amine, created as previously described (*Hill et al., 2012*), 1 Eq of **1a** (0.2 M) and 1.1 Eq of **1b** (Abachemscene) were dissolved in DMF at RT. The reaction was placed on ice. While stirring, 1.3 Eq HOAt (1-Hydroxy-7-azabenzotriazole) and 3 Eq DIEA (N,N-Diisopropylethylamine) were added. After 5 min on ice, 1.3 Eq of EDCI (1-Ethyl-3-(3-dimethylaminopropyl)carbodiimide) was added. The reaction was allowed to stir for 24 hr (letting the ice melt and the reaction slowly come to RT). Reaction was dissolved in ethyl acetate, washed with $NaHCO_3$ and brine, and dried with $Na_2SO_4$. Remaining solvent and DMF were removed via rotovaping and lyophilization. **1** c was deprotected with 30% TFA in DCM (0.2 M **1** c final). Solid **1** c was dissolved in $CH_2Cl_2$ and cooled on ice. TFA was added dropwise until it reached 30% v/v. Reaction was stirred for 1 hr at RT. Toluene was added (to help remove TFA) and the reaction was rotovapped to near dryness. Reaction was dissolved in ethyl acetate, washed with $K_2CO_3$, dried $Na_2SO_4$ and dried via rotovapping and lyophilization. Identity at each step was verified with MS.

### BI2536 functionalization (+CLP-linker)

**Chemical structure 2.** BI2536 functionalization (+CLP-linker).

1 Eq **1d** (0.2 M) and 1.1 Eq of **2a** (Chem Scene) were dissolved in DMF at RT. The reaction was placed on ice. While stirring, 1.3 Eq HOAt and 3 Eq DIEA were added. After 5 min on ice, 1.3 Eq of EDCI was added. The reaction was allowed to stir for 24 hr (letting the ice melt and the reaction slowly come to RT). DMF was removed and **2b** (BI2536-CLP) was purified with HPLC. Identity was verified with MS. [M+H]+ = 817.7 *m/z*.

### MLN8237 functionalization (+CLP-linker)

**Chemical structure 3.** MLN8237 functionalization (+CLP-linker).

1 Eq **1d** (0.2 M) and 1.1 Eq of **3a** (MLN8237) were dissolved in DMF at RT. The reaction was placed on ice. While stirring, 2 Eq HOAt and 3 Eq DIEA were added. After 5 min on ice, 1.2 Eq of EDCI was added. The reaction was allowed to stir for 24 hr (letting the ice melt and the reaction slowly come to RT). DMF was removed and **3b** (MLN8237-CLP) was purified with HPLC. Identity was verified with MS. [M+H]+ = 911.0 *m/z*. **CLP-rhodamine Preparation:** 1 Eq **1a** (0.2 M) and 1 Eq 5(6)-carboxytetra-methylrhodamine N-succinimidyl ester (Thermo Fisher) were dissolved in DMF at RT. While stirring, 3 Eq DIEA were added. The reaction was allowed to stir for 24 hr. DMF was removed and product was purified with HPLC. Identity was verified with MS. $[M+H]^+$ = 676.2 *m/z.*

## Protein expression and purification

$His_6$-SNAP-tag in pMCSG7 (Addgene) was expressed in *Escherichia coli* BL21(DE3) cells in 250 mL LB Miller broth. The evening prior to expression, 5 mL LB Miller broth, containing 50 µg/mL Ampicillin, was inoculated with transformed cells, and they were grown at 37°C overnight. The following day, the starter culture was used to seed 250 mL LB Miller broth in a 500 mL baffle flask. Cells were grown to $OD_{600}$ ~0.3 and the temperature was then reduced to 20°C. Cells were allowed to grow to $OD_{600}$ ~0.8, and then induced with 500 µM isopropyl β-D-thiogalactopyranoside. Induced cells were grown at 20°C overnight. Subsequent purification steps were carried out at 4°C. Cells were spun down at 6500 g, suspended in 10 mL of wash/lysis buffer [50 mM HEPES (pH 7.5), 300 mM NaCl, 20 mM imidazole, and 1 mM phenylmethanesulfonyl fluoride], and lysed via sonication. The lysate was centrifuged at 10000 g for 20 min, and the supernatant was allowed to batch bind with 0.7 mL of Ni-NTA ($Ni^{2+}$-nitrilotriacetate) beads for 60 min. The resin was collected by centrifugation at 500 g for 5 min and washed with 10 mL of wash/lysis buffer. The wash step was repeated three times. The Ni-NTA/$His_6$-SNAP-tag was added to a BioRad purification column, and washing was continued until the wash showed no remaining protein by Bradford. The protein was eluted using ~ 5 mL of elution buffer [50 mM HEPES (pH 7.5), 300 mM NaCl, 200 mM imidazole]. The eluate was dialyzed against 50 mM HEPES (pH 7.5), 200 mM NaCl, 5% glycerol, and 1 mM fresh dithiothreitol (DTT). Protein was aliquoted, flash-frozen in liquid $N_2$, and stored at − 80°C.

## SNAP labeling experiments and pulse-chase labeling assays

### In vitro labeling

50 µM SNAP-tag was incubated with 75 µM CLP-linker-inhibitors (or DMSO alone for control reactions) [2.5% (v/v) final DMSO concentration] in buffer [20 mM Tris-Cl (pH 8), 200 mM NaCl, 1 mM DTT (added fresh)] at 26°C for 1.5 hr. The reactions were purified using Zeba columns (Thermo Fisher) and exchanged into a MS compatible buffer (50 mM NH4HCO3, 0.2% HCO2H). Ratios of unlabeled to labeled protein were determined using Native MS (Thermo Scientific LTQ Orbitrap XL/ Bruker Esquire LC-Ion Trap). **Cellular labeling**: SNAP expressing (dox-induced) cells were treated with SNAP-Cell Fluorescein or SNAP-Cell 647-SiR (NEB) for 30 min in serum free DMEM at 37°C and 5% CO2. Cells were washed one time and incubated in fresh serum free DMEM for 30 min. Cells were fixed and stained as described under 'immunofluorescence'. For pulse-chase labeling experiments, SNAP expressing (dox-induced) cells were treated with DMSO or increasing doses of CLP-BI2536 or CLP-MLN8237 for 1, 2, or 4 hr in serum free DMEM at 37°C and 5% CO2. Cells were washed one time and incubated in fresh serum free DMEM for 30 min. Cells were treated with 3 µM CLP-rhodamine (made in-house) in serum free DMEM for 30 min. Cells were washed one time and incubated in fresh serum free DMEM for 30 min. One wash with PBS was carried out and cells were lysed using immunoblotting protocol. Samples were resolved on an AnykD Criterion TGX Precast Midi Protein Gel (Biorad). Gels were scanned and fluorescence was measured with a GE Typhoon FLA 9000 scanner. Fluorescence measurements and densitometry was performed using NIH ImageJ (Fiji) software. Total SNAP labeling was determined by normalizing fluorescence signal of rhodamine bands to total SNAP protein expression as determined by densitometry.

% SNAPs Labeled by CLP-rhodamine = CLP rhodamine signal/SNAP western blot signal

% SNAPs Labeled by CLP-inhibitor = 100 % SNAPs Labeled by CLP-rhodamine

## In vitro kinase assays

### Plk1 assays

The kinase activity of purified Plk1 (SignalChem) was profiled using Casein as the substrate (0.2 mg/mL). Reactions contained 25 mM MOPS, pH 7.2; 12.5 mM β-glycerol-phosphate; 25 mM $MgCl_2$; 2 mM EGTA; 2 mM $Na_3VO_4$; 2 mM BME (β-mercaptoethanol); and 0.05 mg/mL BSA. Serial dilutions (1:3) of 25X compounds in DMSO [4% (v/v) final concentration in the assay] were used. For SNAP-inhibitor titrations, serial dilutions (1:3) were made in assay buffer, and an equivalent amount of DMSO [4% (v/v) final concentration in the assay] was later added to each well. Reciprocally, in assays where 25X inhibitor was added in DMSO, assay buffer was first added to each well (equivalent to the volume of SNAP-inhibitor in buffer). The last two wells in each row served as control reactions (+Kinase/No Inhibitor; and No Kinase/No Inhibitor) and received DMSO or buffer in place of inhibitor or kinase, respectively. After addition and mixing of the above components, kinase stock dilutions (10 nM final) were added to each well (except for the last well in each row, which served as the No Kinase/No Inhibitor control). Next, ATP-[γ $^{32}$P] (final assay concentration: 0.012 µCi/µl) and unlabeled-ATP (final assay concentration: 40 µM) were added to the reactions. Plk1 was preincubated with ATP-competitive inhibitors and ATP-[γ $^{32}$P]/unlabeled ATP for 30 min. To initiate the reactions, Casein was added. The reactions were incubated for 1 hr at RT. **Aurora A assays:** The kinase activity of purified Aurora A (Invitrogen) was profiled using myelin basic protein (MBP) as the substrate (0.2 mg/mL). Reactions contained 30 mM HEPES, pH 7.5; 10 mM $MgCl_2$; 0.6 mM EGTA; 2 mM $Na_3VO_4$; 2 mM BME; and 0.05 mg/mL BSA. Serial dilutions (1:3) of 25X compounds in DMSO [4% (v/v) final concentration in the assay] were used. For SNAP-inhibitor titrations, serial dilutions (1:3) were made in assay buffer, and an equivalent amount of DMSO [4% (v/v) final concentration in the assay] was later added to each well. Reciprocally, in assays where 25X inhibitor was added in DMSO, assay buffer was first added to each well (equivalent to the volume of SNAP-inhibitor in buffer). The last two wells in each row served as control reactions (+Kinase/No Inhibitor; and No Kinase/No Inhibitor) and received DMSO or buffer in place of inhibitor or kinase, respectively. After addition and mixing of the above components, kinase stock dilutions (15 nM final) were added to each well (except for the last well in each row, which served as the No Kinase/No Inhibitor control). To initiate the reactions, ATP-[γ $^{32}$P] (final assay concentration: 0.006 µCi/µl) was added to the reactions. The reactions were incubated for 4 hr at RT. **All assays:** Assays were quenched by spotting 4.6 µL of each reaction mixture onto phosphocellulose membranes (Reaction Biology). The membranes were subjected to three sequential washes in 0.5% phosphoric acid for 10 min, dried, and exposed overnight to a phosphor screen (GE Healthcare). Blots were scanned using a phosphor scanner (GE Typhoon FLA 9000). Raw data were processed with the GraphPad Prism software package (V5.0a) using the One site - Fit logIC$_{50}$ function for curve fitting. Spots were quantified using ImageQuant. The kinase activity was first determined to be linear under assay conditions for Plk1 experiments at 10 nM Plk1 and at 15 nM Aurora A for Aurora A experiments, before conducting inhibitor titrations.

## Drug treatments

For induction of SNAP expression cells were treated for 48–72 hr in FBS-supplemented DMEM with 1 µg/mL (for SNAP-PACT) or 4 µg/mL (for SNAP-Mis12) doxycycline hyclate (Sigma-Aldrich) prior to inhibitor treatments. For degradation assays, cells were dox-induced for 72 hr after which doxycycline was washed out (cells were incubated in normal media). At selected time point plates were collected, cells were washed once with PBS, plates were dried quickly, and frozen at −80°C until lysis. For nocodazole synchronization experiments, dox-induced cells were treated for 16 hr with nocodazole and 4 hr with nocodazole plus DMSO, 250 nM CLP-BI2536, or 100 nM CLP-MLN8237. Cells were washed once with PBS, collected via mitotic shake-off, and spun at 2000 rpm for 5 min at 4°C. Supernatants were discarded and pellets were kept for lysis. All lysates were prepared as described under 'immunoblotting'. For fixed cell experiments, both dox-induced and non-induced cells were grown on 1.5 poly-D-lysine coated coverslips (neuVitro) for at least 16 hr in complete DMEM and then treated with DMSO or CLP-compounds in serum-free DMEM for 1–4 hr. For washout experiments (pT210-Plk1 1 hr washout and γ-tubulin data), cells were incubated in serum-free DMEM without inhibitors for an additional 1 hr. Cells were washed once with PBS prior to fixation. For live-imaging experiments, cells were treated with CLP-compounds for 18 hr (see 'microscopy' for more details).

## Zebrafish studies

Zebrafish were bred and embryos were collected. Embryos were injected with EMTB-3xGFP (100 pg mRNA or 20 pg pCS2 plasmid construct), SNAP-Cell Fluorescein (300 µM final embryo concentration), and/or CLP-BI2536 (250 nM final embryo concentration), and/or SNAP-PACT active or dead (200 pg mRNA) at the 2 cell stage using a microinjector (Warner Instruments PLI-100A) with a Kanatec magnetic base (MB-B), and a micromanipulator (Marzhauser Wetzlar MM33). Embryos were raised at 30°C until ~50% epiboly, at which point they were imaged on stereoscope, mounted in 2% agarose for live confocal imaging, or fixed using 4% paraformaldehyde + 0.5% Triton-X overnight at 4°C. Embryos were dechorionated in PBST (phosphate buffered saline + 0.1% Tween-20) and incubated in DAPI solution (1 µg/mL in PBS) for 2 hr at room temperature. Fixed embryos were then mounted in 2% agarose and imaged on confocal microscope.

## Immunoblotting

Cells were lysed in RIPA lysis buffer (50 mM Tris HCl pH 7.4, 1% Triton X-100, 0.5% Sodium Deoxycholate, 0.1% SDS, 50 mM NaF, 120 mM NaCl, 5 mM β-glycerophosphate) supplemented with protease and phosphatase inhibitors (1 mM benzamidine, 1 mM AEBSF, 2 µg/mL leupeptin, 100 nM microcystin-LR). Lysed samples were boiled for 5 min at 95°C in NuPAGE LDS Sample Buffer 4X (Thermo Fisher) + 5% BME (Sigma-Aldrich). Protein concentration was determined using a Pierce BCA Protein Assay Kit (Thermo Fisher). Samples were resolved on Bolt 4–12% Bis-Tris Plus Gels (Invitrogen) or AnykD Criterion TGX Precast Midi Protein Gel (Biorad). Proteins were transferred to nitrocellulose for immunoblotting and probed with Anti-SNAP-tag rabbit antibody (NEB) and Anti-GAPDH−HRP mouse mAb, (Novus). Detection was achieved with a HRP-conjugated rabbit secondary antibody (GE Healthcare) followed by enhanced chemiluminescence with SuperSignal West Dura Extended Duration Substrate (Thermo Fisher). Densitometry was performed using NIH ImageJ (Fiji) software.

## Immunofluorescence

Cells grown on 1.5 poly-D-lysine coated coverslips (neuVitro) for at least 16 hr were fixed for 10 min in 4% paraformaldehyde in PBS or in ice-cold methanol. Cells were permeabilized and blocked in PBS with 0.5% Triton X-100% and 1% BSA (PBSAT) for 30 min. Primary antibodies were diluted in PBSAT and cells were stained for 1 hr. Secondary antibodies conjugated to Alexa Fluor dyes (Invitrogen) were diluted in PBSAT and applied for 1 hr. Staining with FITC-tubulin antibodies and/or DAPI staining always followed secondary incubation step and was carried out for 10–45 min in PBSAT. Washes (quick on and off) with PBSAT were carried out 10X between antibody and/or dye incubation steps and prior to mounting. Coverslips were mounted on slides using ProLong Diamond Antifade Mountant (Life Technologies).

## Spindle classification measurements

To de-identify cell type and dox-treatment conditions and allow for blinded analysis of mitotic spindle differences, all identifying information on microscope slides was masked by a third-party individual. All mitotic cells within a coverslip were classified as either having normal bipolar, abnormal bipolar, or monopolar spindles based on morphology of the DNA and microtubules.

## Microscopy
### Fixed cell

Super-resolution 3D-SIM images were acquired on a Deltavision OMX V4 (GE Healthcare) system equipped with a 60x/1.42 NA PlanApo oil immersion lens (Olympus), 405-, 488-, 568-, and 642 nm solid-state lasers and sCMOS cameras (pco.edge). For SIM, 15 images per optical slice (3 angles and five phases) were acquired. For both SIM and widefield (conventional) acquisitions, image stacks of 2.5–7 um with 0.125 um optical thick z-sections were acquired using immersion oil with a refractive index 1.516 or 1.518. Z-stacks were generated using the SNAP-PACT or SNAP-Mis12 channel to define the upper and lower regions of the plane with a 0.5 µM step size. SIM images were reconstructed using Wiener filter settings of 0.003 and optical transfer functions measured specifically for each channel with SoftWoRx software (GE Healthcare) to obtain super-resolution images with a twofold increase in resolution both axially and laterally. Images from different color channels were

registered using parameters generated from a gold grid registration slide (GE Healthcare) and Soft-WoRx. Widefield images were deconvolved using SoftWoRx. **Live-cell**: For LoKI experiments cells were first induced with doxycycline for 48–72 hr prior to transfection. For all time-lapse experiments cells were reverse transfected with GFP-H2B plasmid and plated onto μ-Slide 4 Well Glass Bottom: # 1.5H (170 μm + /- 5 μm) D 263 M Schott glass (Ibidi) in complete DMEM. Transient transfections were performed using TransIT-LT1 reagent (Mirus) with Opti-MEM (Life Technologies) media. The next day all cells were treated with 2 mM thymidine for 24 hr. The following day thymidine was washed out and after 4 hours cells were incubated with DMSO or CLP-inhibitors (LoKI experiments) or with no reagents (WT/KO Gravin experiments) in serum-free FluoroBrite DMEM. Time-lapse images were acquired on a Keyence BZ-X710 microscope using a 10X objective with 25% transmitted light and 100% aperture stop, with 1/60 s exposure for 488 channel. Images were captured every 5 min for 18 hr. **Zebrafish imaging**: Images were acquired on a Leica DMi8 (Leica, Bannockburn, IL) equipped with a Crest Optics X-light v2 Confocal Unit spinning disk, an 89 North – LDI laser with a Photometrics Prime-95B camera using a Nikon 40 × 1.15 N.A. Lamda S LWD objective. Stereoscope images were acquired on a Leica M165 FC stereoscope with a DFC9000 GT camera and a PLANAPO 10X objective.

## Image analysis

Maximum intensity projections from z-stack images were generated using SoftWoRx (GE Healthcare) or NIH ImageJ (Fiji) software. All immunofluorescence signal measures were carried out using Fiji software. Sum slice 32-bit Tiff projections were generated from z-stack images for analysis of immunofluorescence at centrosomes. For kinetochore measurements the ImageJ 'SubtractMeasuredBackground' macro was first applied and sum slice 32-bit Tiff projections were generated. For centrosomes measurements, the oval selection tool in Fiji was used to draw a circle (ROI) around the centrosome in the 568 (SNAP-PACT) channel. The area of the circle remained consistent for all measurements and all replicates of an experiment. Measurements were taken in the 647 channel (which contained pT210-Plk1, Total Plk1, pT288-AurA, or γ-tubulin) using the predefined centrosome ROI. Using the measure function in Fiji, with 'Area' and 'Raw Integrated Density' predefined as measurements, values were determined for each centrosome and for an arbitrarily selected background region. The raw integrated density was recorded for each centrosome and the background. The average raw integrated density for the centrosomes was determined by adding together the raw integrated densities for each centrosome in a cell and dividing that value by 2. The integrated density for the background was subtracted from the average centrosome integrated density to yield a background-subtracted average integrated density signal for a centrosome. If the signal value was negative (signal at centrosome was lower than at background) the value was replaced with a 0. For kinetochore measurements, the selection tool in Fiji was used to draw an arbitrary region (ROI) around the kinetochore in the 405 (ACA, centromeric DNA) channel or in the 568 (SNAP-Mis12) channel. Measurements were taken in the 647 channel (which contained pS69-Hec1) using the predefined kinetochore ROI. Using the measure function in Fiji, with 'Area' and 'Raw Integrated Density' predefined as measurements, values were determined for each kinetochore. The raw integrated density was recorded for each kinetochore. For both centrosome and kinetochore experiments and average was calculated for each control and experimental condition. To do this the normalized average integrated densities were added together and divided by the total number of cells for that condition. This yielded a value that represents the background-normalized average integrated density at a centrosome or at the kinetochore for a particular condition. Values for drug-treated cells were then normalized to their respective DMSO-treated control. Integrated intensity surface plots were generated from sum-slice 32-bit Tiff projections of representative images using the 3D Surface Plot function in Fiji software. Maximum intensity heat maps were generated from maximum projection representative images using the 3D Surface Plot function with Fire LUT in Fiji software. **Zebrafish three-dimensional renderings**: Three-dimensional renderings were created using Imaris software (Bitplane). Individual mitotic cells were isolated and assigned a new color channel using the 'Surfaces' function to create a surface rendering. Surface renderings were created through the use of the Isoline function, where regions of individual mitotic cells were isolated based on intensity. Completed surface renderings were then merged and masked to create a channel that encompassed the mitotic cells of each embryo.

## Statistical analysis

Statistics were performed using an unpaired two-tailed Student's t-test in GraphPad Prism software. All values are reported as mean ± standard error of the mean (s.e.m) with p-values less than 0.05 considered statistically significant. Number of independent experiments (N) and number of individual points over several experiments (n) are presented. For γ-tubulin experiments a ROUT (Q = 1%) outlier test was performed and two values were removed prior to performing an unpaired Student's t-test.

## Sample size and replicates

The sample size was not statistically determined. Where applicable, n > 15 independent measurements were conducted across N ≥ 3 independent experiments. For doxycycline removal experiments (*Figure 2—figure supplement 1E*, *Figure 6—figure supplement 1B*) at least 2–3 independent experiments were conducted per time point.

## Acknowledgements

We thank members of the Scott Lab for critical discussions, Jennifer DeLuca (Colorado State University) for the pS69-Hec1 antibody, Patrina Pellett (GE Healthcare) for technical help with super-resolution imaging techniques, Juan-Jesus Vicente (Wordeman Laboratory, UW) for help with experimental design, data analysis, and thoughtful discussion.

## Additional information

### Funding

| Funder | Grant reference number | Author |
|---|---|---|
| National Institutes of Health | 5R01DK105542 | John D Scott |
| National Institutes of Health | 1R01DK119192-01 | John D Scott |
| National Institutes of Health | R01GM127621 | Heidi Hehnly |
| National Institutes of Health | R01GM086858 | Dustin J Maly |
| National Science Foundation | DGE-1762114 | Paula J Bucko |
| National Institutes of Health | T32GM008268 | Chloe K Lombard |

The funders had no role in study design, data collection and interpretation, or the decision to submit the work for publication.

### Author contributions

Paula J Bucko, Conceptualization, Data curation, Formal analysis, Supervision, Validation, Investigation, Visualization, Methodology, Writing—original draft, Project administration, Writing—review and editing; Chloe K Lombard, Data curation, Formal analysis, Investigation, Methodology; Lindsay Rathbun, Data curation, Formal analysis, Investigation; Irvin Garcia, Data curation, Formal analysis, Validation, Investigation; Akansha Bhat, Validation, Investigation; Linda Wordeman, Conceptualization, Resources, Formal analysis; F Donelson Smith, Conceptualization, Supervision; Dustin J Maly, Conceptualization, Supervision, Funding acquisition, Methodology; Heidi Hehnly, Data curation, Supervision, Funding acquisition, Methodology; John D Scott, Conceptualization, Supervision, Funding acquisition, Visualization, Writing—original draft, Project administration, Writing—review and editing

### Author ORCIDs

Paula J Bucko (iD) https://orcid.org/0000-0002-0889-5465
F Donelson Smith (iD) http://orcid.org/0000-0002-8080-7589
Dustin J Maly (iD) http://orcid.org/0000-0003-0094-0177
Heidi Hehnly (iD) http://orcid.org/0000-0001-6660-5254
John D Scott (iD) https://orcid.org/0000-0002-0367-8146

## Ethics

Animal experimentation: Animal experimentation: This study was performed in strict accordance with the recommendations in the Guide for the Care and Use of Laboratory Animals of the National Institutes of Health. All of the animals were handled according to approved Institutional Animal Care and Use Committee (IACUC) protocols (#18-006) of the Syracuse University.

## Decision letter and Author response

Decision letter https://doi.org/10.7554/eLife.52220.sa1
Author response https://doi.org/10.7554/eLife.52220.sa2

## Additional files

### Supplementary files

• Transparent reporting form

### Data availability

Raw analysis and quantification files are provided as source data. Source data files have been provided for Figures 2, Figure 3, Figure 4, Figure 6 and Figure 2—figure supplement 2, Figure 3—figure supplement 1 and Figure 6—figure supplement 1.

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

## Appendix 1

# Supplementary information

## Clarification of terminology

In this paper, LoKI refers to the general method/tool that can be utilized for studying local kinase action. In addition, LoKI-on refers to the presence of a platform that enables local kinase inhibition while LoKI-off serves as a control. For simplicity, we refer to cells that localize CLP substrates at defined subcellular locations as LoKI-on and those that globally distribute these substrates throughout the cell as LoKI-off (reflecting 'traditional' drug treatment paradigms). The former is always accomplished by doxycycline-induced expression of catalytically active (wildtype) SNAP moieties that can covalently bind CLP-compounds. Conversely, we have two ways to accomplish LoKI-off. In some experiments LoKI-off refers to cells that express SNAP tags containing a single amino acid change (C144A) in SNAP. This mutation results in a catalytically-inactive enzyme that is incapable of binding CLP. LoKI-off can also refer to non-doxycycline-induced wildtype SNAP cells. These cells lack SNAP expression entirely and thus are likewise incapable of localizing CLP substrates. Both our doxycycline-induced mutant SNAP and non-induced wildtype SNAP cells serve to mimic traditional forms of drug delivery in which small-molecule inhibitors are free to diffuse globally throughout the cell.

## Components of LoKI platforms

All LoKI constructs used in this study contain a fluorescent reporter protein, two SNAP domains, and a subcellular localization sequence. For the centrosome and kinetochore-localized constructs an N-terminal mCherry reporter protein and a C-terminal localization sequence (PACT or Mis12, respectively) were fused to the double-SNAP moiety. The second SNAP has a varied codon sequence to simplify the PCR-amplification process. For plasma membrane and mitochondrial-localized constructs a C-terminal eGFP reporter protein and an N-terminal localization sequence (AKAP79 or dAKAP1, respectively) were fused to the SNAP domains. Again, the second SNAP had a varied codon sequence.

