## [Decision Letter]

**Acceptance summary:**

This paper explores the idea of combining AKAP targeting domains and SNAP-tagging technologies to inhibit kinases at specific subcellular locations. The authors demonstrate the feasibility of this platform by targeting AurA and Plk1 inhibitors to centromeres or kinetochores. This strategy provides evidence clarifying the actions of centrosome- or kinetochore-localized AurA and Plk1. The strategy demonstrated in this paper is novel and exciting and will be of broad interest to the kinase field.

**Decision letter after peer review:**

Thank you for submitting your article "Subcellular drug targeting illuminates local kinase action" for consideration by *eLife*. Your article has been reviewed by three peer reviewers, including Ivan Dikic as the Reviewing Editor and Reviewer #1, and the evaluation has been overseen by Jonathan Cooper as the Senior Editor. The following individual involved in review of your submission has agreed to reveal their identity: Vincent S Tagliabracci (Reviewer #2).

The reviewers have discussed the reviews with one another and the Reviewing Editor has drafted this decision to help you prepare a revised submission.

Summary:

This paper explores the idea of combining AKAP targeting domains and SNAP-tagging technologies to inhibit kinases at specific subcellular locations. The authors demonstrate the feasibility of this platform by targeting AurA and Plk1 inhibitors to centromeres or kinetochores. This strategy provides evidence clarifying the actions of centrosome- or kinetochore-localized AurA and Plk1. Overall, this is an interesting chemical biology study addressing the issue of local inhibition of protein kinases. The authors have gone a long way to exemplify a solution in cells and a model organism. The strategy demonstrated in this paper is exciting and will be of broad interest to the kinase field. We recommend publication in *eLife*, but have a couple of comments that the authors should consider when preparing a revised manuscript.

Essential revisions:

1) Combined AurA and Plk1 inhibitions were done using a dual SNAP conjugation moiety (Figure 4F). In theory, it is hard to ensure that both CLP-MLN8237 and CLP-BI2536 would be recruited by a dual SNAP moiety. If technically feasible, it would be good if the authors can use orthogonal tagging strategies in tandem, such as SNAP-tag plus CLIP-tag, to demonstrate this point. Also, the authors should show the effect of either single drug in Figure 4H, which should be included as additional controls.

2) The authors claim that the LoKI platform can achieve local kinase inhibition. While the authors do provide evidence that 1) pS69-Hec1 signal at centrosomes was not affected if CLP-MLN8237 was directed to kinetochores (Figure 6G, H; representative images shown but better quantified) and 2) pS69-Hec1 signal at kinetochores was not affected if CLP-MLN8237 was directed to centrosomes (Figure 6—figure supplement 1E), it would be good if the authors can perform off-target assessment using TPX2 phosphorylation in the condition that CLP-MLN8237 is directed to centrosomes. TPX2 phosphorylation by AurA occurs at spindle microtubules in the neighbourhood of centrosomes, and thus TPX2 phosphorylation presumably would be a better index to tell how local/leaky the LoKI platform is. This is also related to the fact that in vitro kinase assays showed that CLP-inhibitor conjugates work to inhibit their cognate kinase at around 100nM, which is c. 10-fold worse than the unconjugated inhibitor. When raising the CLP-drug concentration then inhibition becomes independent of the SNAP targeting protein. The authors should discuss these points in the manuscript.

3) One general concern is that the limitations of the approach are not discussed, giving the sense of an ultimate solution which is not the case. It would have been instructive to have had discussion of the timing problem – 4h is a long period of drug exposure and somewhat undermines the need for acute local intervention, particularly in these cell cycle associated events. Additionally, it is evident that effects are observable in relatively narrow windows in order to distinguish between global inhibition and local inhibition. This does not undermine the study but appropriate discussion would give direction to those following on. It is recommended that the authors comment and discuss the advance in technology from the work by Gower et al., 2016. Some comment is needed to address that Gravin knockdown leads to increased Plk1 mobility and CEP215 phosphorylation (10.1091/mbc.E17-08-0524). This relates to the principle of insulation of signalling (i.e. by sequestration). This emphasizes the need to be able to inhibit locally rather than just remove/mutate the scaffold responsible.

---

## [Author Response]

Essential revisions:1) Combined AurA and Plk1 inhibitions were done using a dual SNAP conjugation moiety (Figure 4F). In theory, it is hard to ensure that both CLP-MLN8237 and CLP-BI2536 would be recruited by a dual SNAP moiety. If technically feasible, it would be good if the authors can use orthogonal tagging strategies in tandem, such as SNAP-tag plus CLIP-tag, to demonstrate this point. Also, the authors should show the effect of either single drug in Figure 4H, which should be included as additional controls.

This raises an excellent point. There are three parts to our response.

a) We determined that the concentration of inhibitor required to bind ~50% of the available SNAP-tags in a 4 hour incubation window as 250 nM for CLP-BI2536 and 100 nM for CLP-MLN8237 (Figure 2F and Figure 4C). We agree that having one CLP-linked inhibitor targeted to SNAP while the other inhibitor derivatized to interact with a different self-labeling enzyme would be a logical extension of our work. However, in addition to synthesizing the new drug (compatible for interaction with CLIP-tag) and generating new constructs, we would need to validate that the modified drug still inhibits its target, is cell permeable, and establish a concentration regime that provides localized inhibition. Thus, while developing orthogonally-tagged versions may be technically feasible, this venture would require many months of work. In addition, there is no guarantee that cystozine-benzyl-BI2536 or cystozine-benzyl-MLN8237 adducts would retain their efficacy.

b) In an effort to respond to the reviewers' suggestion we have carried out mitotic duration experiments that examine the effect of either drug alone. The inclusion of this data in a new supplementary figure (Figure 4—figure supplement 3) demonstrates the effect of individual inhibitors on mitotic duration. This material is discussed in the subsection “Combined Plk1 and AurA suppression at centrosomes more profoundly delays mitosis than global kinase inhibition”. An important conclusion from these added studies is that targeting single inhibitors delays mitosis to a lesser extent than combination drug-targeting.

c) In accordance with point 3 (see below) we have expanded the Discussion noting that “we hope that future work will advance on our strategy and provide a system that utilizes multiple self-labeling enzymes to deliver distinct inhibitors to the same location”.

2) The authors claim that the LoKI platform can achieve local kinase inhibition. While the authors do provide evidence that 1) pS69-Hec1 signal at centrosomes was not affected if CLP-MLN8237 was directed to kinetochores (Figure 6G, H; representative images shown but better quantified) and 2) pS69-Hec1 signal at kinetochores was not affected if CLP-MLN8237 was directed to centrosomes (Figure 6—figure supplement 1E), it would be good if the authors can perform off-target assessment using TPX2 phosphorylation in the condition that CLP-MLN8237 is directed to centrosomes. TPX2 phosphorylation by AurA occurs at spindle microtubules in the neighbourhood of centrosomes, and thus TPX2 phosphorylation presumably would be a better index to tell how local/leaky the LoKI platform is. This is also related to the fact that in vitro kinase assays showed that CLP-inhibitor conjugates work to inhibit their cognate kinase at around 100nM, which is c. 10-fold worse than the unconjugated inhibitor. When raising the CLP-drug concentration then inhibition becomes independent of the SNAP targeting protein. The authors should discuss these points in the manuscript.

We provide three responses to point 2.

a) As per the suggestion we now provide quantification in support of our statement in the manuscript. We demonstrate that pS69-Hec1 at centrosomes remains unaffected by CLP-MLN8237 treatment over a range of concentrations. We hope that this new data in Figure 6—figure supplement 2A will help support our original claim. These new results are introduced in the last paragraph of the subsection “Targeting CLP-MLN8237 to kinetochores reveals that AurA-mediated Hec1 phosphorylation is a local event”.

b)The reviewers recommended that we look at TPX2 phosphorylation as a measure of AurA activity at spindles to assess the leakiness of the LoKI platform. While this is a thoughtful suggestion, it is not technically possible to conduct these studies with existing reagents. Previous work from Fu et al., 2015 has demonstrated that TPX2 phosphorylation (as assessed by gel mobility shift) is reduced when mitotic cell lysates are treated with AurA inhibitors in vitro. Although phos-tag analyses is a means to indirectly evaluate phosphorylation status of TPX2, there are currently no reagents to selectively evaluate AurA-dependent phosphorylation of TPX2 inside cells. Moreover, since TPX2 is enriched at centrosomes during early mitosis (Eibes et al., 2018) it would be unclear whether loss of TPX2 phosphorylation (assessed by phos-tag) was due to inhibition of AurA at centrosomes, spindles, or both. In order to respond to the reviewers’ suggestions we have expanded on these points in the Discussion. In the modified manuscript we envision that future advancements of this platform would include photo-caged inhibitors that are inert until they are ready to be released after binding to SNAP at the site of desired inhibition. This would provide another level of control and more strictly define the range of inhibitor action. This statement is introduced in the fifth paragraph of the Discussion.

c)With all due respect, we wonder if the reviewers may have misinterpreted our in vitro kinase data. We observed an IC_50_ of <9.5 nM for CLP-MLN8237 and an IC_50_ of <11 nM when CLP-MLN8237 is bound to SNAP (Figure 4B, Figure 4—figure supplement 1B). In cells, we find that high concentrations of CLP-drug (100 nM) are required to produce equivalent effects as non-derivatized MLN8237 (10nM). For comparison please see Figure 4E versus Figure 4—figure supplement 2B. We postulate that this is an issue with reduced cell permeability of certain CLP-drug conjugates which necessitates their use at approximately 10-fold higher concentrations inside cells. These points have been more clearly articulated in the Discussion (fourth paragraph).

3) One general concern is that the limitations of the approach are not discussed, giving the sense of an ultimate solution which is not the case. It would have been instructive to have had discussion of the timing problem – 4h is a long period of drug exposure and somewhat undermines the need for acute local intervention, particularly in these cell cycle associated events. Additionally, it is evident that effects are observable in relatively narrow windows in order to distinguish between global inhibition and local inhibition. This does not undermine the study but appropriate discussion would give direction to those following on. It is recommended that the authors comment and discuss the advance in technology from the work by Gower et al., 2016. Some comment is needed to address that Gravin knockdown leads to increased Plk1 mobility and CEP215 phosphorylation (10.1091/mbc.E17-08-0524). This relates to the principle of insulation of signalling (i.e. by sequestration). This emphasizes the need to be able to inhibit locally rather than just remove/mutate the scaffold responsible.

We have expanded the Discussion to highlight the limitations of our novel chemical-biology strategy. This includes commentary on the timing and concentration required to achieve sufficient targeting of our inhibitors and the issue of altered cell-permeability with our drug adducts (Discussion, fourth paragraph). We discuss how generating photo-caged inhibitor drugs would provide an additional level of control and allow us to better decipher the effects of global versus local kinase inhibition (Discussion, fifth paragraph). Additionally, we incorporate discussion on the advancements of our technology in relation to work by Gower et al., 2016 (Discussion, first paragraph). Finally, we cite Colicino et al., 2018 and provide a commentary on the need to develop novel drug-targeting strategies to better probe the local roles of Gravin-anchored kinases (Discussion, first paragraph).